# Intracellular acidification is a hallmark of thymineless death in *E. coli*

**Alexandra Ketcham**[1,2,3], **Lydia Freddolino**[2,3¤], **Saeed Tavazoie**[1,2,3*]

**1** Department of Biological Sciences, Columbia University, New York, New York, United States of America,
**2** Department of Systems Biology, Columbia University, New York, New York, United States of America,
**3** Department of Biochemistry and Molecular Biophysics, Columbia University, New York, New York, United States of America

¤ Current address: Department of Biological Chemistry, University of Michigan Medical School, Ann Arbor, Michigan, United States of America
* st2744@columbia.edu

**Data availability statement:** The authors confirm that all data underlying the findings

## Abstract

Thymidine starvation causes rapid cell death. This enigmatic process known as thymineless death (TLD) is the underlying killing mechanism of diverse antimicrobial and antineoplastic drugs. Despite decades of investigation, we still lack a mechanistic understanding of the causal sequence of events that culminate in TLD. Here, we used a diverse set of unbiased approaches to systematically determine the genetic and regulatory underpinnings of TLD in *Escherichia coli*. In addition to discovering novel genes in previously implicated pathways, our studies revealed a critical and previously unknown role for intracellular acidification in TLD. We observed that a decrease in cytoplasmic pH is a robust early event in TLD across different genetic backgrounds. Furthermore, we show that acidification is a causal event in the death process, as chemical and genetic perturbations that increase intracellular pH substantially reduce killing. We also observe a decrease in intracellular pH in response to exposure to the antibiotic gentamicin, suggesting that intracellular acidification may be a common mechanistic step in the bactericidal effects of other antibiotics.

## Author summary

Thymineless death was discovered in *E. coli* but the killing phenomenon is widespread—from eukaryotic microbes to human cells. The mechanism of killing is essential to the mode of action of several clinically important drugs, yet the causal sequence of events that lead to cell death remains poorly understood. We undertook three systems biology approaches to probe the underlying mechanisms of thymineless death in *E. coli* and discovered dozens of novel genes that modulate survival in previously unknown pathways. The function of many of the genes pointed to pH homeostasis as a critical factor. Our detailed studies revealed that intracellular acidification occurs during the thymidine starvation process and experimental manipulation of pH causes dramatic effects on survival. We also observed that a genetic perturbation that increases intracellular pH affects survival after treatment with the antibiotic gentamicin. Indeed, gentamicin exposure causes

are fully available without restriction. All relevant data are either within the paper or its Supporting Information files, while raw and processed high-throughput sequencing data used in this study are available from the NCBI Gene Expression Omnibus under accessions GSE147758 and GSE147760.

**Funding:** This work was supported by awards from the NIH: R01AI077562 (to S.T.); R35GM128637 (to L.F.); 1F31GM108419 (to A.K.). The funders had no role in study design, data collection and analysis, decision to publish, or preparation of the manuscript.

**Competing interests:** The authors have declared that no competing interests exist.

a decrease in pH, suggesting that acidification may play a broader role in the bactericidal effects of other antibiotics.

## Introduction

Deoxythymidine triphosphate (dTTP) is one of the four nucleoside triphosphates required for DNA replication. Its *de novo* biosynthesis requires the action of thymidylate synthetase, encoded by *thyA* in *E. coli*. When a *thyA* mutant is starved of exogenous thymidine, it rapidly dies [1,2]. This killing phenomenon, TLD, was subsequently shown to occur in other bacteria, yeast, and human cells [3]. Thymidine starvation-induced killing is closely linked to the mode of action of several clinically important antibacterial, antineoplastic, and antimalarial drugs such as methotrexate [4], trimethoprim [5], and 5-fluorodeoxyuridine [6]. Resistance has been observed for all of these drugs [7–9]. Thus, a better understanding of the toxic phenomenon is likely to have broad clinical implications.

Although DNA synthesis is sharply inhibited during TLD, there is still an increase in DNA content in thymidine-starved *E. coli* cells [2,10–12] and a correlation exists between the extent of lethality and the number of replication forks per cell [13]. TLD is reduced by mutations in or near the origin of replication (*oriC*) that interfere with the required transcription step for initiating replication [14,15]. Inactivating the major DNA replication and repair proteins also modulates TLD kinetics, and the proteins involved fall into two categories: those with protective roles and those that enhance killing [3].

Early researchers in the field reported that stopping respiration (by oxygen removal) stops killing [16]. Several years later, it was discovered that *E. coli c*ells deficient in the respiration protein cytochrome oxidase, encoded by *cydA*, are protected from TLD in rich medium [17]. Recently it was found that the enhanced survival of respiration mutants is associated with lower accumulation of endogenous reactive oxygen species (ROS) levels, suggesting that it is the buildup of ROS and resultant damage to DNA that kill cells during thymidine starvation [18].

Although replication initiation, DNA damage, and ROS accumulation have all been observed in bacteria during thymidine starvation and implicated in the killing process, the precise sequence of events from thymidine starvation, DNA damage, and ROS accumulation to death remains unknown. Another central question is whether additional pathways also contribute to the killing process.

In this work, we undertook three complementary systems biology approaches to systematically probe the underlying mechanisms of TLD. We quantified the contribution of every non-essential gene in the genome to TLD, evolved strains that exhibit extreme TLD-resistance, and probed the transcriptional responses of TLD-sensitive and resistant strains. Genes involved in DNA replication and repair, electron transport chain and/or ROS accumulation, and pH homeostasis showed concordant effects across independent approaches and the tested genetic backgrounds. We observed that intracellular pH (pHi) strongly correlates with survival at both early and late time points, and that cytoplasmic acidification precedes ROS accumulation. Consistent with a causal role in death, experimental modulation of pH significantly affected survival. Finally, we provide evidence that intracellular acidification also occurs upon exposure to the antibiotic gentamicin, suggesting that it may be a common early step in the lethality of other bactericidal perturbations.

## Results

### Tn-seq profiling reveals a role for pH modulators in TLD

A saturated Tn5 transposon insertion library was generated in a *thyA⁻* strain to ensure that no cell had more than one transposon insertion and to ensure that the library included dozens of insertions to all nonessential genes in the genome. This library was subsequently selected in thymidine-free media and insertions were mapped and sequenced at different time points to look for changes in survival caused by gene disruptions (Figs 1A and S1A). In this case, each sequencing read mapped to a gene at a particular time point during thymidine (T) starvation represents the presence of a cell with a disruption in that gene in the total population. A survival score for each gene was defined as the log₂ fold change (LFC) of sequencing read

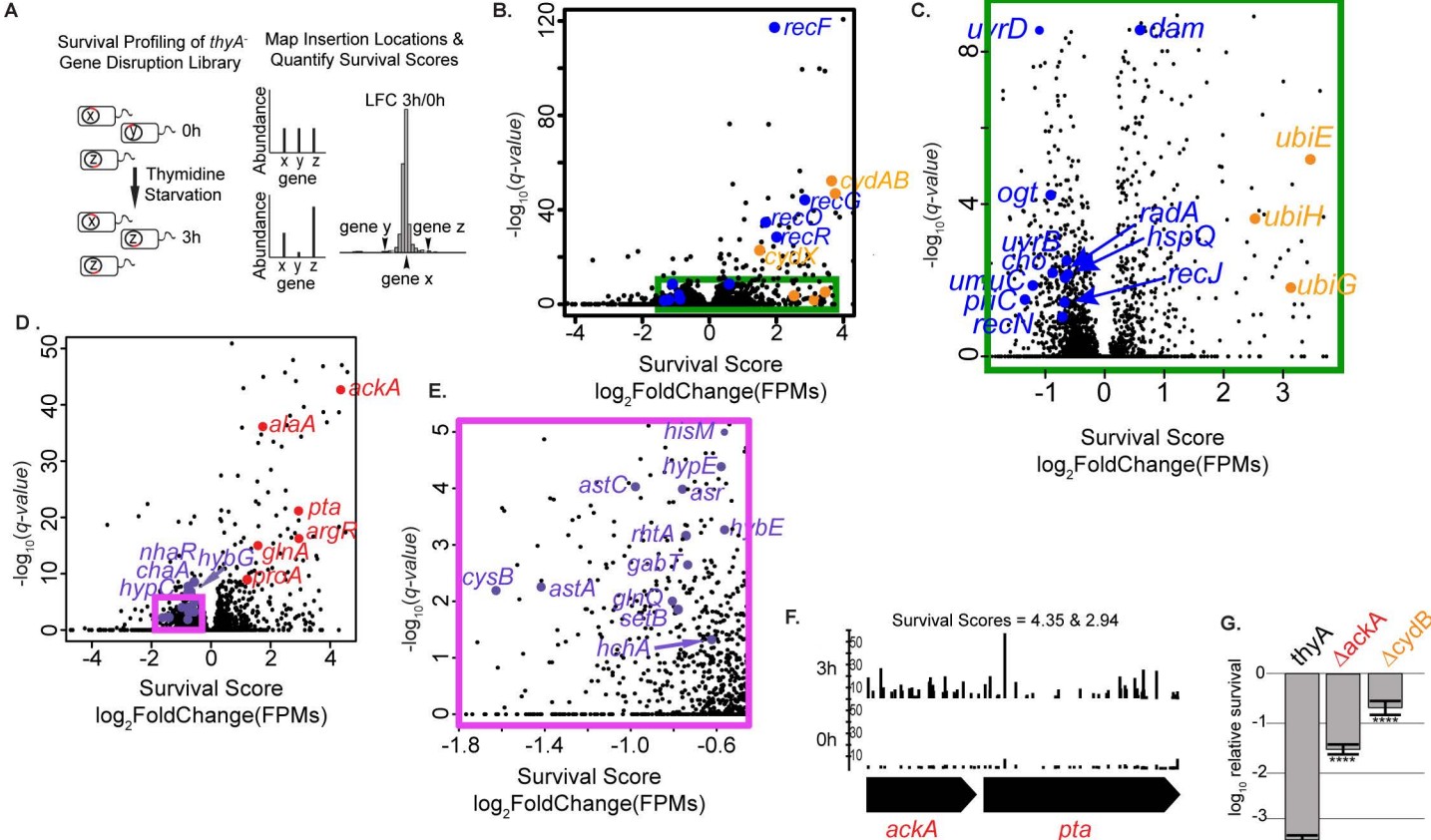

**Fig 1. Systematic genetic survey of TLD by Tn-seq reveals a novel role for pH homeostasis.** (A) A transposon insertion library was generated in the MG1655 *thyA⁻* strain and starved for thymidine. DNA adjacent to each insertion was amplified and sequenced at the beginning of the selection and at 3h. For each gene, the survival score is the log₂ fold change of normalized reads at 3h/0h. (B-C) Volcano plot of survival score and significance showing genes in previously known pathways in color. Here and in the rest of this work, orange represents genes associated with respiration/ROS and blue represents genes involved in DNA replication/repair. Significance was calculated using an exact rate ratio test (using the rateratio.test R package) with a Benjamini & Yekutieli correction [103]. (C) is an inset for the area in the rectangle pictured in (B). (D-E) Volcano plot of survival score and significance showing in color genes from candidate lists that modulate cytoplasmic pH. Disruptions in genes involved in proton influx, or in lowering substrates needed for deacidification, have positive survival scores (shown in red). Conversely, disruptions in genes whose products are directly involved in proton consumption or producing substrates required for deacidification systems have negative survival scores (in light blue). (E) is an inset for the area in the rectangle pictured in (D). (F) Frequency of transposon insertions along the length of two newly identified TLD contributors at 0h and at 3h thymidine starvation. The enhancements at 3h are visualized using the Integrated Genome Browser [104]. The frequencies of each insertion site are shown in read counts per million. (G) Enhanced survival of validated genes. Candidates were validated by transferring Keio collection knockout alleles into MG1655 *thyA⁻*, and assessing their survival at 3h of thymidine starvation. All death assays, unless otherwise stated, were performed at 37°C. Relative survival was measured for at least three independent experiments, with error bars representing standard error of the mean. *p*-values for all death assays were calculated using a Welch t-test. * P<0.05, ** P<0.01, *** P < 0.001, **** P<0.0001. Here and in the rest of this work, red color represents genes in the acetate dissimilation pathway.

transposon footprints per million (FPM) at 3h versus FPM at 0h (S1 Table). Cells with disruptions in genes that hurt survival during T-starvation will be more abundant in the population after 3 hours (e.g. reads to that gene will increase in frequency from 0h to 3h and thus it will have a positive survival score). Cells with disruptions in genes that help survival during T-starvation will be less abundant in the population after 3h (e.g. reads to that gene will decrease and thus that gene will have a negative survival score).

Upon filtering the normalized survival scores for inconsistencies in survival using a no-outgrowth control (see Methods), Tn-seq analysis revealed 212 significant genes in which insertions exacerbate killing and 175 significant genes in which insertions alleviate killing (S2 and S3 Tables and S1B and S1C Fig). In order to provide another layer of statistical rigor we added an additional test for high-confidence hits. In order to qualify as a high-confidence hit from the Tn-seq screen, the following criteria must all be met: (1) The gene must have at least two separate transposon insertion locations each of which shows an independently significant change in insertion frequency between the conditions being compared (q<0.1 using the R prop.test function), and (2) all significant insertions in that gene must have the same direction of fitness effects. These additional criteria help prevent the possibility of PCR jackpotting or other unusual events with single clonal lineages from altering our overall conclusions. Overall, ~60% of genes initially identified in the Tn-seq screen also passed our additional multiple-insertion screening criterion. High confidence hits are noted as such in Tables 1 and 2 below and in S1 Table.

Of the genes previously known to alleviate or sensitize TLD when disrupted, all of the survival scores showing statistically significant effects were consistent with prior observations (Fig 1B, 1C and S4 Table and S1D and S1E Fig). Thus, previously implicated genes have survival scores concordant with previously existing expectations in our screen. Of the 21 genes in Table 1 with significant survival effects from our screen that fall into previously implicated pathways, 10 have not been previously implicated in TLD: *cho*, *cydX*, *dam*, *hspQ*, *ogt*, *priC*, *radA*, *recN*, *ubiEH*.

By choosing genes with significant survival scores above a threshold and cross referencing with the no-outgrowth control (see Methods), we generated a list of 52 candidate genes with positive survival scores for which deletion is expected to be beneficial for TLD survival (S5 Table). From this list, we generated 12 in-frame knockouts in the MG1655 *thyA*⁻ background, taking care to select genes across different pathways and GO Term categories. Nine of the 12

**Table 1. Genes with significant effects from survival profiling candidate lists that fall into the previously known pathways of DNA replication/repair and respiration.**

| Gene | Survival Score |
|---|---|
| *cho** | -1.21 |
| *cydA*B*X* | 3.76, 3.65, 1.50 |
| *dam* | 0.59 |
| *hspQ** | -0.67 |
| *ogt** | -0.90 |
| *priC** | -1.0 |
| *radA* | -0.62 |
| *recF*G*J*N*O*R** | 1.95, 2.85, -0.64, -0.68, 1.68, 2.01 |
| *ubiEGH** | 3.46, 3.14, 2.53 |
| *umuC** | -1.34 |
| *uvrB** | -0.88 |
| *uvrD** | -1.1 |

*indicates high confidence hits

**Table 2. Genes from survival profiling candidate lists that fall in the newly identified pathway of pH homeostasis.**

| Pathway | Gene | Survival Score |
|---|---|---|
| acetate dissimilation | *ackA** | 4.35 |
| glutamate metabolism | *alaA** | 1.74 |
| negative regulation of arginine biosynthesis | *argR** | 2.95 |
| acid shock protein | *asr** | -0.76 |
| arginine catabolic process to glutamate | *astA** | -1.42 |
| arginine catabolic process to glutamate | *astC** | -0.97 |
| sodium ion: proton antiporter | *chaA** | -0.76 |
| controls arginine-dependent acid resistance | *cysB** | -1.63 |
| glutamate formation | *gabT** | -0.73 |
| glutamate metabolism | *glnA** | 1.57 |
| L-glutamine transport | *glnQ** | -0.81 |
| response to acidic pH; mutant more sensitive to acid stress | *hchA* | -0.61 |
| lysine arginine uptake | *hisM* | -0.56 |
| hydrogenase 2-specific chaperone | *hybE** | -0.57 |
| hydrogenase 3 maturation protein | *hypC** | -0.78 |
| **Na**$^+$/**H**$^+$ **a**ntiporter **R**egulator | *nhaR** | -0.57 |
| proline biosynthesis from glutamate | *proA** | 1.21 |
| acetate dissimilation | *pta** | 2.94 |
| proton antiporter; pH homeostasis | *rhtA** | -0.74 |
| proton antiporter | *setB** | -0.78 |

*indicates high confidence hits

knockouts that we generated and tested showed significantly increased survival from the *thyA*$^-$ at 3h T-starvation (S1F Fig), showing empirically that our screen had good validation accuracy and allowed for the identification of previously unknown genes involved in TLD survival.

A common theme among many of the genes identified by our Tn-seq survey, which do not belong to already known pathways, was involvement in pH homeostasis. We observed that disruptions in genes that produce or import H$^+$ into the cytoplasm, or that lower levels of substrates needed for deacidification systems, enhance survival during thymidine starvation, and disruptions in genes that consume protons, or produce substrates needed for deacidification systems, exacerbate killing (Fig 1D and 1E and Table 2).

In order to test whether the identified gene categories of interest were enriched in our survival profiling datasets without relying on GO term annotations (which may not exactly align with the processes of interest), we defined sets of genes involved in response to acid stress, DNA damage, or oxidative stress, using data from the Fitness Browser database [19]. We used genes affecting survival of benzoic acid, nalidixic acid, or sodium chlorite to define our acid stress, DNA damage, and oxidative stress categories, respectively (see Methods for details). We found strong and significant associations of genes that were important in all three categories among those with significant survival scores in our transposon library experiments, with log$_2$ fold enrichments of genes important for fitness in the noted stress condition (relative to background) of +1.74 ($p$ = 1.084e-8; Chi squared test) for acid stress, +0.71 ($p$ = 0.006 Chi squared test) for DNA damage stress, and +0.85 ($p$ = 0.026 Chi squared test) for oxidative stress. Thus, we observed significant enrichments for functionally relevant genes from all three categories considered here amongst our transposon library hits.

Insertions in genes *cydA* and *cydB* led to the highest survival scores in the list of candidates in previously known pathways (S1G Fig). The novel candidate contributor, *ackA*, encoding

acetate kinase of the acetate production and excretion pathway, had the highest survival score of the candidate genes with a role in pHi modulation (Fig 1F). Acetate production and excretion play prominent roles in intracellular acidification since acetate, once excreted, can freely permeate back across the cell's membrane into the cytoplasm where it dissociates delivering protons and lowering the pH [20,21]. Inactivation of the *ackA* gene and the related gene in this pathway, *pta*, has been shown to cause constitutive extreme-acid resistance as well as lower acidification in the medium [22,23]. The effects of *cydB* and *ackA* on TLD were each validated by transferring the corresponding Keio knockout allele to the MG1655 *thyA*⁻ strain and assessing their survival during thymidine starvation. Clean deletions in both of these genes showed significant enhancements in survival compared to the *thyA*⁻ strain background (Fig 1G).

## Laboratory evolution of extreme TLD resistance

A complementary and unbiased way to probe the *E. coli* genome for genes involved in TLD is through long-term laboratory evolution [24]. Clones of the *thyA*⁻ strain in two genetic backgrounds (MG1655 and MDS42 [25]) were selected independently for increased survival in supplemented defined media in limiting amounts of thymidine (0.2µg/mL) and propagated for ~50 days, with one transfer per day into fresh medium (Fig 2A). For the majority of the transfers (43/50), only 2-fold dilutions were performed due to the robust killing. While each parental strain shows 2–3 orders of magnitude of death at 3h thymidine starvation, the best surviving isolate from each background shows very little death (Fig 2B). In fact, it takes the best surviving evolved isolates in each background 18-50h to suffer the same magnitude of loss as that suffered by the parental strain at 3h (Fig 2C).

Whole-genome sequencing revealed that the number of mutations in the two evolved isolates ranged from 15 in the MDS42 evolved strain to 69 in the MG1655 evolved strain (Tables 3 and S6). Three of the mutations occurred in loci that were previously characterized in the TLD literature: *recO* [26–28] in the MG1655 evolved strain, and both *recJ* [27–30], and *oriC* [15] in the MDS42 evolved strain.

The *oriC* and *recJ* mutations in the MDS42 evolved isolate are not causing all of this strain's resistance. Another isolate derived from the same evolved population does not survive as well during thymidine starvation (Fig 2D). Whole-genome sequencing on both sibling isolates shows that they share all mutations except that the better surviving isolate has an *atpF* early stop codon. The sibling that survives less well has wild type *atpF*. The *atpF* gene encodes subunit *b* of the proton-transporting (F0) portion of membrane bound ATP synthase. This protein complex catalyzes the synthesis of ATP using the energy of an electrochemical ion gradient by moving protons into the cell. It has not been previously implicated in TLD, but in principle is connected both to pH homeostasis and aerobic respiration.

To quantify the degree to which *atpF* plays a role in TLD resistance, a *ΔatpF* allele was transferred into the MG1655 *thyA*⁻ strain. Deletion of *atpF* significantly enhances the survival of the parental strain (Fig 2E) without slowing the growth rate in our defined media (Table 4, S2 Fig, and methods). Mutations in (or upstream of) the operon for the proton-transport-coupled ATP-synthase genes occur across both independently evolved isolates (Tables 3 and S6). A *recO* deletion was made in order to compare its survival with the novel mutants. Combining *ΔatpF* and *ΔrecO* significantly enhances the survival relative to either in isolation (Fig 2E). This increase in survival cannot be attributed to slower replication, as *ΔatpFΔrecO* has a faster growth rate than either mutation in isolation (Table 4 and S2 Fig).

In addition to recurring mutations in proton-transport-coupled ATP-synthase, both independently evolved isolates have mutations in putrescine biosynthesis genes, including a non-synonymous mutation in *speB*, whose transposon disruption significantly enhanced survival (S1H Fig and Table 3). *speB* encodes agmatinase, a putrescine biosynthesis enzyme.

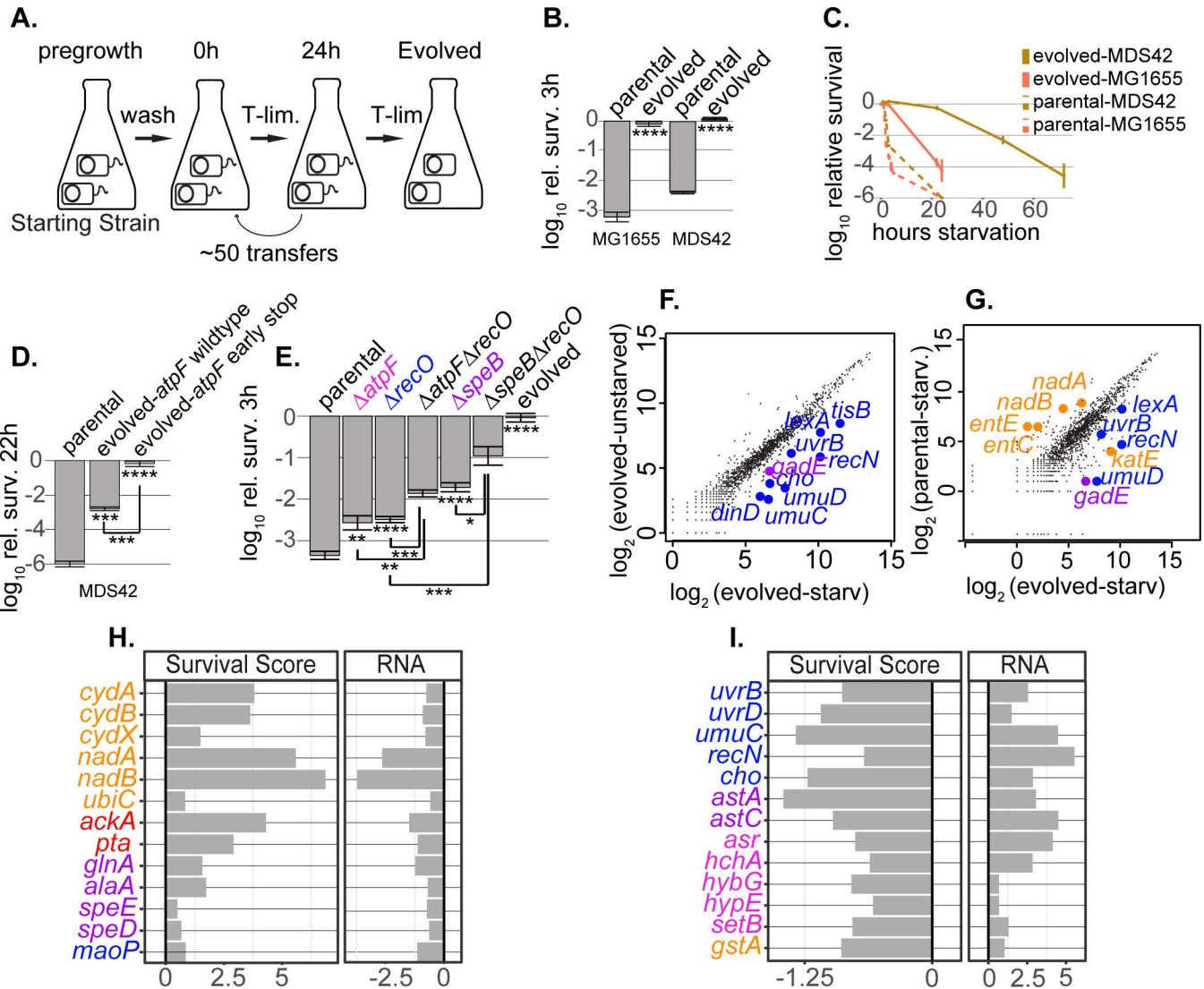

**Fig 2. Genetic and transcriptional evidence for the role of pH homeostasis in laboratory evolution of extreme TLD-resistance.** (A) Experimental setup of laboratory evolution. Clones of *thyA⁻* strains in the MG1655 and MDS42 backgrounds were selected independently in 0.2μg/mL thymidine. After roughly 50 transfers, isolates were characterized for enhanced survival. (B) Short-term death assay for evolved isolates and parents at 3h thymidine starvation. The stars in Fig 2 correspond to the *p*-values as specified in Fig 1 and are calculated using a Welch t-test. (C) Long-term death assay for evolved isolates and parents. The points are averages from independent experiments, with error bars representing standard error of the mean. (D-E) Survival of TLD-sensitive versus various TLD-resistant strains. The stars at the bottom of a bar represent the *p*-values calculated for that strain compared to the parental strain. The stars under brackets represent the *p*-values calculated between the strains bracketed. (D) *atpF* early stop contributes to extreme TLD resistance. Two sibling isolates from the same evolved population share all mutations except an *atpF* early stop codon (the strain that survives less well has wild type *atpF*). The three strains shown are in the MDS42 background. (E) Clean deletion validations at 3h thymidine starvation. All knockouts are in the MG1655 *thyA⁻* background. Here and in the rest of this work, purple represents genes in putrescine / glutamate / arginine metabolism, and pink represents genes involved in proton translocation (or sequestration) systems. (F-G) Transcriptome profiling of TLD-sensitive and TLD-resistant strains. RNA was extracted and sequenced 30 minutes into thymidine starvation for the evolved strain in the MDS42 background and its parent strain. RNA was also extracted and sequenced in the unstarved condition. Select genes showing significant differential expression are shown in color. Rockhopper [99,100] uses a negative binomial distribution to estimate the uncertainties in the read counts. The *p*-values are FDR-corrected using the Benjamini-Hochberg procedure [101]. Genes in the Replication/ Repair, ROS, and pH homeostasis pathways are shown in color. For comprehensive tables of the LFC of mRNA expression for the parental and evolved strains, see S7 and S8 Tables. (F) Gene expression of the evolved strain during starvation versus unstarved. (G) Gene expression of evolved versus parental strain during thymidine starvation. (H-I), Select genes with concordant effects across MG1655 and MDS42 backgrounds and across experimental approaches. In each, the left-hand column shows survival scores from the survival profiling experiment in the MG1655 strain and the right-hand column shows LFC of TPM in the evolved versus parental in the MDS42 background during thymidine starvation. (H) Genes that exacerbate TLD. The left-hand-column shows positive survival scores and the right-hand column shows down-regulation of RNA expression levels of the evolved strain vs. parental during starvation. (I) Genes that alleviate TLD. The left-hand-column shows negative survival scores and the right-hand column shows up-regulation of RNA expression levels of the evolved strain versus parental during starvation. See S9 Table for the full list of genes showing concordant effects.

**Table 3. Mutations in the TLD-resistant evolved isolate in the MDS42 background.**

| Gene | Mutation | Gene Function |
|---|---|---|
| *atpF* | Q85* (G→A) | F0 sector of membrane-bound ATP synthase, subunit b |
| *gidA* ← / ← *mioC* | intergenic (-239/+140) (T→A) | *mioC* and *gidA* are the well-conserved genes surrounding the *E. coli* replication origin, *oriC* |
| *glnH* | L5F (TTA→TTT) | glutamine transporter subunit |
| *glnH* | V4V (GTA→GTT) | glutamine transporter subunit |
| *lsrF* | I153L (A→C) | predicted aldolase |
| *ptsH* | A20T (GCC→ACC) | phosphohistidinoprotein-hexose phosphotransferase component of PTS system (Hpr) |
| *recJ* | Δ1 bp coding (1191/1734 nt) | ssDNA exonuclease, 5' —> 3'-specific |
| *speB* | I59F (T→A) | agmatinase |
| *tdcD* | F146S (TTC→TCC) | propionate kinase/acetate kinase C, anaerobic |
| *ycjW* | T4A (ACT→GCT) | predicted DNA-binding transcriptional regulator involved in the bacterial stringent response. |
| *ygeG* → / → *ygeH* | intergenic (+197/-138) (A→T) | predicted chaperone/predicted transcriptional regulator |
| *yihU* ← / → *yihV* | intergenic (-102/-66) (G→T) | predicted oxidoreductase with NAD(P)-binding Rossmann-fold domain/predicted sugar kinase |
| *yjeP* | A853T (GCG→ACG) | predicted mechanosensitive channel |
| *yjeP* | V794A (GTC→GCC) | predicted mechanosensitive channel |
| *yraK* | I237N | part of putative chaperone-usher fimbrial operon |

Putrescine has many diverse functions in the cell, including a role in pH homeostasis [31,32]. Its metabolism is intricately linked to that of glutamate and arginine. Together these inter-linked processes comprise three of the cell's amino-acid-dependent decarboxylase acid resistance systems, which play critical roles in deacidification in *E. coli* [33]. When a *speB* deletion was transferred to the *thyA*⁻ MG1655 strain, it showed significant enhanced survival compared to its parent (Fig 2E). Furthermore, a *ΔspeB* mutant combined with *ΔrecO* shows significantly enhanced survival compared to either strain in isolation (Fig 2E).

**Table 4. Growth rates of strains in rich defined media.**

| Strain | Average doublings per hour, 37°C |
|---|---|
| MG1655 *thyA*⁺ | 2.09 |
| MG1655 *thyA*⁺ + Arg | 1.96 |
| MG1655 *thyA*⁻ | 2.26 |
| MG1655 *thyA*⁻ + Arg | 2.07 |
| evolved MG1655 | 1.82 |
| MDS42 *thyA*⁺ | 2.01 |
| MDS42 *thyA*⁻ | 2.29 |
| Evolved MDS42 | 1.51 |
| sibling of Evolved MDS42 | 2.19 |
| ΔackA | 1.81 |
| ΔatpF | 2.57 |
| ΔatpFΔackA | 1.06 |
| ΔatpFΔrecO | 2.61 |
| ΔcydB | 1.77 |
| ΔcydBΔackA | 1.96 |
| ΔrecO | 2.28 |
| ΔrecOΔackA | 1.79 |
| ΔspeB | 2.27 |
| ΔspeBΔackA | 1.39 |
| ΔspeBΔrecO | 2.21 |

It is important to note that in principle, slowed bacterial growth might be expected to account for some of the effects on survival observed in both the evolved strains and targeted gene deletions considered here. However, apart from strains containing *ΔackA* alleles (which will be discussed further below), in general we did not observe a strong correlation between growth rate and TLD survival (S2 Fig)–for example, deletions of *recO*, *speB*, or *atpF* all strongly enhanced survival without affecting growth rate, and evolved strains show no to minor loss of growth rate while at the same time showing massively improved survival of TLD.

## The evolved TLD-resistant strain shows adapted transcriptional responses to DNA damage, ROS, and acid stress

In order to better understand how the evolved strain may have rewired its transcriptional output to enhance survival, RNA sequencing was performed on the evolved and parental strains in both thymidine-starved and unstarved conditions (S3A Fig and S7 Table). Strains in the MDS42 background were used, due to the lower number of accumulated (and potentially non-TLD related) mutations in the evolved strain. Whereas both the parental and evolved strain showed a transcriptional response to thymidine starvation, the evolved strain's response was more pronounced (S3B and S3C Fig).

In Fig 2F and 2G we highlight selected genes that show strong differential regulation between the evolved starved population and either the evolved unstarved (F) or parental starved (G) populations. Several SOS-induced genes were significantly upregulated in the evolved strain in multiple comparisons, suggesting that these genes may have protective roles during TLD (Figs 2F, 2G and S4B and S4C, in blue and S8 Table). Previous results have shown that, during TLD, the SOS response is induced [26,28,34,35]. Our observations support a protective role for a subset of SOS-induced genes during the early stages of thymidine starvation.

In addition to the marked DNA-damage response, we saw evidence that the TLD-resistant strains have acquired improved capacity to cope with ROS. For example, *katE*, one of the two catalases in *E. coli* which serve as the primary scavengers for hydrogen peroxide [36,37], is significantly upregulated in the evolved versus parental in both starved and unstarved conditions (Figs 2G and S4A and S4B). The *de novo* NAD biosynthesis genes *nadA* and *nadB* are significantly downregulated in the evolved versus parental comparison during thymidine starvation (Figs 2G and S4B, in orange). These genes have not, to our knowledge, been previously implicated in TLD survival. However, the product of *nadB* was reported to be a predominant source of endogenous hydrogen peroxide under aerobic conditions [38]. The product of *nadA* has an oxygen-sensitive iron-sulfur cluster that is required for its activity [39,40]. Iron-sulfur clusters are known targets and sources of ROS due to the resulting displacement of iron from the cofactor and the intimate connection between labile iron and ROS generation [41,42]. Iron-acquisition genes *entE* and *entC* are also significantly downregulated in the evolved versus parental comparison during starvation (Figs 2G and S4B). Conversely, *entE* and *entF* are significantly upregulated in the parental starved versus unstarved (S4D Fig).

These data suggest that the TLD-resistant strain shows improved responses to both DNA damage and ROS accumulation during thymidine starvation. The significant upregulation of the gene *gadE* in the evolved strain, across multiple comparisons, suggests that TLD-resistant strains may also be responding to acid stress during thymidine starvation (Figs 2F, 2G and S4B and S4C). *gadE* encodes the most important regulatory component of the glutamate-dependent acid response (GAD) system [43]. The product of *gadE* controls the transcription of *gadA* and *gadB*, genes involved in pH homeostasis [44]. For a comprehensive list of the RNA expression levels in the parental and evolved strains, see S7 and S8 Tables.

## Concordant genetic and transcriptional evidence for the involvement of pH homeostasis genes in TLD

We hypothesized that genes whose products alleviate killing during thymidine starvation would not only have a negative survival score when disrupted, but also that their expression would be upregulated in the TLD-resistant strain. Conversely, genes whose products exacerbate killing during TLD were expected not only to show positive survival scores when disrupted, but that their expression would be down-regulated in the TLD-resistant strain. Transcript counts from the transcription profiling experiment were normalized to transcripts per million (TPMs) in order to allow for more direct comparison with the survival profiling data.

A search for genes that show concordant effects in genetic and transcriptional responses yielded several lists of candidates (S9–S11 Tables). We saw an enrichment in genes that are upregulated and for which KO is harmful during thymine starvation in all four RNA-seq comparisons (S5 Fig). We do not see the analogous enrichment for genes that are downregulated and for which the KO is helpful. One explanation is that due to our stringent thresholds for excluding genes with very low read counts, our assay is less sensitive in picking up downregulated genes than upregulated genes.

All three lists contain dozens of genes involved in DNA replication and repair, the electron transport chain and/or ROS accumulation, and pH homeostasis (Figs 2H and 2I and S6A and S6B). Most of these genes have not, to the best of our knowledge, been previously implicated in TLD.

Two examples of genes in the pH homeostasis pathway that show concordant effects across experimental approaches are *glnA* (Fig 2H) and *setB* (Fig 2I). Disruptions in *glnA* (which consumes glutamate, needed for the GAD system) help survival during T-starvation with a +1.57 survival score ($\log_2$ fold change, or LFC, of FPMs). Consistently, the transcription profiling showed that expression of this gene is downregulated in the resistant strain compared to the sensitive strain in the starved condition, with a -1.2 LFC of TPMs. On the other hand, disruptions in *setB*, which encodes a proton antiport, involved in exporting protons out of the cell (just as the GAD system does), hurts survival with a -0.78 survival score. Transcription profiling showed that expression of this gene is upregulated in the resistant strain compared to the sensitive strain in the starved condition, with a +1.3 LFC of TPMs.

Another example of a pH homeostasis gene which shows concordant effects across experimental approaches and appears to play a critical role in TLD is *asr*, the acid shock protein (Figs 2I and S6A and S6B). Asr expression is induced under acidic conditions [45] and plays a role in survival in low pH [46]. We find that disruptions in this gene hurt survival during thymidine starvation, with a survival score of -0.76 from the Tn-seq profiling experiment. Its mRNA expression is upregulated in the starved resistant strain in every condition we tested (for the resistant vs sensitive strain in the starved condition the LFC of TPMs is +4.14 (Fig 2I); for the resistant vs sensitive strain in the unstarved condition the LFC of TPMs is +9.40 (S6B); for the resistant strain in the starved condition vs the resistant strain unstarved the LFC of TPMs is +0.65 (S6A). There is no difference in expression in the starved sensitive strain vs the unstarved sensitive strain (0.09 LFC of TPMs) (see S7 Table). This may help explain the sensitive strain's poor survival during T-starvation.

## Cells undergoing TLD show intracellular acidification before evidence of ROS accumulation

In order to test whether pH plays a role in TLD, we utilized two independent pH-sensitive dyes, BCECF-AM and pHrodo Green AM, to monitor pHi dynamics by flow cytometry. The

absorbance of the fluorescein derivative 2′,7′-Bis-(2-carboxyethyl)-5-(and-6-) carboxyfluorescein 4 (BCECF) is sensitive to pH; its pKa makes it ideal for reporting pH$_i$ at physiological conditions [47]. pHrodo is a newer rhodamine-based probe that is weakly fluorescent at neutral pH but interacts with hydrogen ions resulting in fluorescence (ThermoFisher Scientific). The addition of acetoxymethyl (−AM) groups to both probes makes them membrane permeable to facilitate loading into cells. Once inside, hydrolysis of the acetyl ester linkage by enzymatic cleavage regenerates the less permeable compound, which is retained within the intracellular space whereupon its fluorescence intensity acts as an indicator of pHi.

We also used Peroxy Orange 1, a cell permeable boronate compound specifically activated by intracellular $H_2O_2$, to monitor ROS generation [48,49]. Upon reaction with $H_2O_2$, a highly fluorescent product is released and can be assayed by fluorescence imaging. In order to determine the most informative time points to monitor cellular redox and pH states, we carried out high temporal resolution kill curves for the most sensitive strains and chose the first time point at which we began to see death (1.5h), as well as 3h, the time point at which we gauged survival for most of our death assays. S7 Fig shows kill curves for the most sensitive strains before, at, and after the chosen flow cytometry time points. All of the flow cytometry data has been normalized for cell size and refined size-dependent blank values, to rule out artifactual effects (see Methods).

The *thyA⁻* strains in both backgrounds showed significant ROS accumulation as well as intracellular acidification at 3h thymidine starvation (Figs 3A–3F and S8A–S8C and Methods). These effects are starvation-specific; they were not observed in the *thyA⁺* strain grown in thymidine-free media nor in the parental strain in thymidine-supplemented media (Figs 3A, 3C and 3E).

While the *thyA⁻* strain showed significant increases in acidification compared to the *thyA⁺* strain at 1.5h starvation for both pH dyes, it did not show evidence of ROS accumulation at this earlier time point, suggesting that acidification may precede ROS accumulation (Fig 3B, 3D and 3F). In order to ensure that the lack of observable ROS accumulation at the earlier time point was not the result of a delay in dye internalization or activation, the *thyA⁻* strain was visualized at 1.5h starvation after a 30-minute incubation with 1mM hydrogen peroxide. Since there is modest death at this early time point (Fig 3G at 1.5h), we assumed if ROS is involved it would be within the range of the lowest concentrations for mode one killing, which is due to DNA damage: 1 to 3 mM $H_2O_2$ [50]. We tested the lowest concentration within this range as in [51]. A significant elevation in ROS was observed for these samples (S9 Fig).

Unlike the TLD-sensitive strains, neither of the two evolved strains exhibited intracellular acidification nor showed signs of ROS accumulation at 3h thymidine starvation (Figs 3A, 3C, 3E and S8A–S8C); likewise, the individual knockout strains that showed significant enhanced survival (Fig 3G) were also assessed for evidence of both ROS accumulation and intracellular acidification during thymidine starvation using flow cytometry, and showed general reductions particularly at 3h (Fig 3H–3M).

## Sorting an isogenic population of thymidine-starved cells into high and low cytoplasmic acidification demonstrates increased survival in the low acidification population

In order to test more directly whether the cytoplasmic acidification changes we observe during thymidine starvation are associated with changes in survival, an isogenic population of *thyA⁻* cells was starved of thymidine and stained with both pH sensitive dyes (in parallel experiments). At 1.5h, which was the earliest time point at which we saw death and at which we observed significant acidification changes without accompanying significant changes in

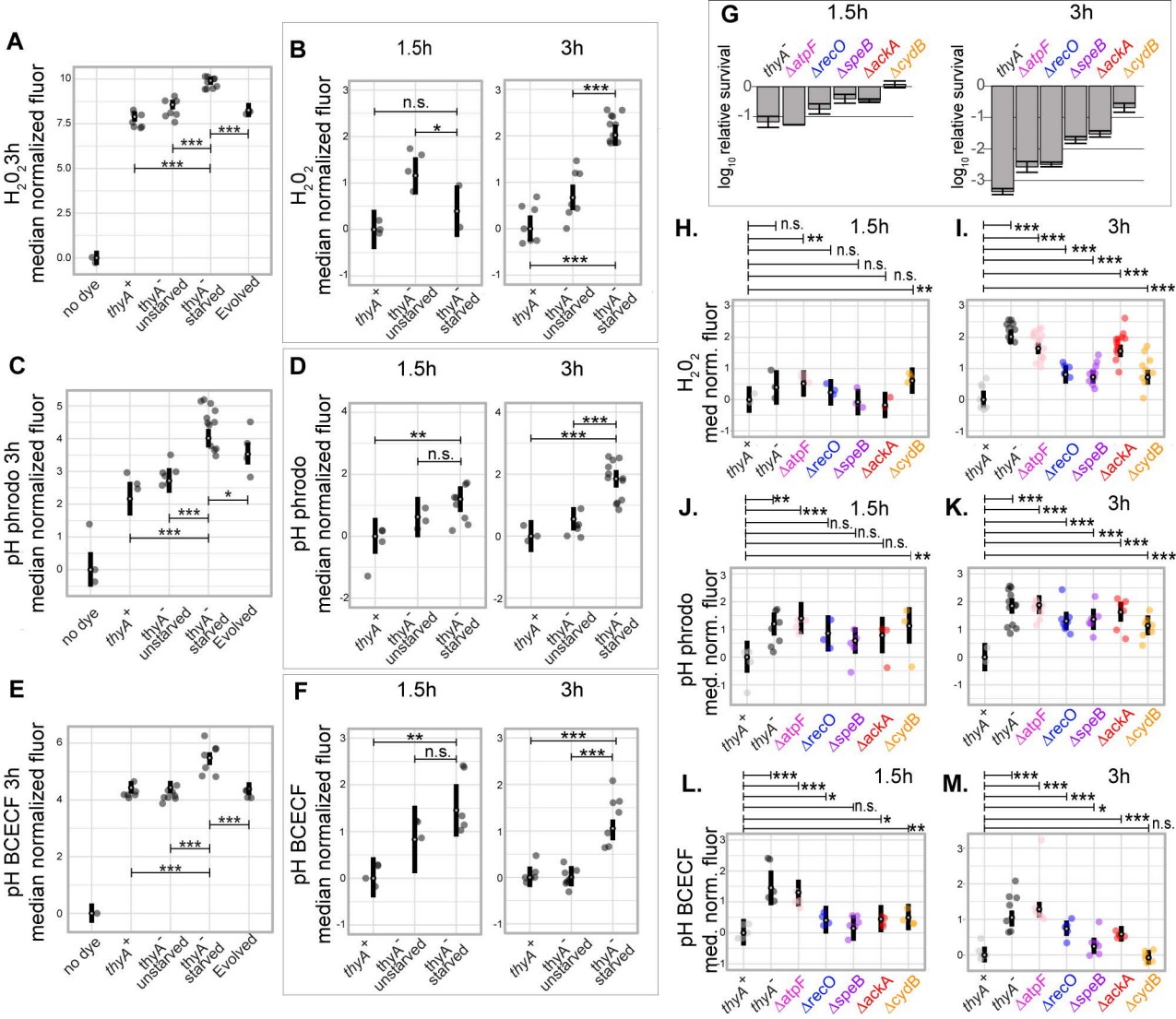

**Fig 3. Cells undergoing TLD show intracellular acidification followed by ROS accumulation, with the degree of survival correlated with pH.** Fluorescence data after accounting for cell size/shape (see Methods). Fluorescence was measured for each independent experiment using flow cytometry; in all flow cytometry figures unless otherwise noted, the large point for each condition shows the fitted effect for that biological condition, with error bars showing a 95% confidence interval. In addition, the median for each biological replicate is shown as a smaller translucent point (note that the shown biological replicates do not reflect corrections due to the random effect terms of the model, and thus show the actual biological variability of the experiment prior to model regularization). All values are offset by the fitted value for the first condition shown, which is thus centered on zero. Stars show significance tests based on the mixed effects model: * P<0.05, ** P<0.01, *** P < 0.001, **** P<0.0001. (A-F) Adjusted fluorescence for *thyA+*, parental, and evolved strains in the MG1655 background. All strains are in thymidine-free media except for those marked "unstarved". (A-B) Adjusted fluorescence for strains dyed for $H_2O_2$ using Peroxy Orange. (A) Adjusted fluorescence measured at 3h. (B) Adjusted fluorescence measured at 1.5h and 3h. (C-D) Adjusted fluorescence measured for strains dyed for pHi using pHrodo Green. (C) Adjusted fluorescence measured at 3h. (D) Adjusted fluorescence measured at 1.5h and 3h. (E-F) Adjusted fluorescence measured for strains dyed for pHi using BCECF-AM. (E) Adjusted fluorescence measured at 3h. (F) Adjusted fluorescence measured at 1.5h and 3h. (G) Survival of the parental strain and knockouts in the MG1655 background at 1.5h and 3h thymidine starvation. (H-M) All strains shown are in thymidine-free media and in the MG1655 background. (H-I) Adjusted fluorescence for strains stained for $H_2O_2$ using Peroxy Orange. (H) Adjusted fluorescence at 1.5h thymidine starvation. (I) Adjusted fluorescence at 3h thymidine starvation. (J-K) Adjusted fluorescence for cells stained for pHi using pHrodo Green. (J) Adjusted fluorescence at 1.5h and (K) Adjusted fluorescence at 3h. (L-M) Adjusted fluorescence for cells stained for pHi using BCECF-AM. (L) Adjusted fluorescence at 1.5h. (M) Adjusted fluorescence at 3h.

ROS accumulation (see above), we sorted cells into the top ~10% and bottom ~10% of the fluorescence distribution, gating stringently so as to only sort cells that are the same size, and then plated them onto high thymidine plates to assess survival differences. Comparing the rates of survivors in the low vs. high fluorescent populations, for both pH-sensitive dyes, we see that there is significantly increased survival in the low acidification sorted cells compared to analogous sorts in the no dye controls (S10A Fig); our model indicates posterior probabilities for higher survival of the low-acidification cells to be 96% (based on BCECF staining) and 99% (based on Phrodo green staining). Thus, even at a very early timepoint, we observe that decreased intracellular pH is associated with worse survival during thymineless death. S10B Fig confirms that the dyes do not affect survival on their own. These data provide additional evidence of a causative link between cytoplasmic acidification and survival changes during thymidine starvation.

## Manipulations that raise the pHi during thymidine starvation increase survival

Several of the genes that showed concordant effects across experimental approaches and the MG1655 and MDS42 backgrounds modulate intracellular levels of amino acids needed for the cell's acid resistance decarboxylation systems. One of these amino acids, arginine, is a precursor of putrescine, and reductions in putrescine synthesis (such as those created by a *speB* knockout) have been observed to co-occur with increases in intracellular arginine concentration [52]. It is thus possible that one way in which the putrescine biosynthesis knockout enhances survival is by raising intracellular arginine levels. Shifts towards arginine accumulation would result in more robust intracellular deacidification, as the decarboxylation of arginine is one of the more powerful amino acid-dependent acid resistance systems in *E. coli* [53].

In order to test the direct involvement of pHi on TLD, L-arginine was added to growth media to increase the pHi and acid resistance [53,54], and to see if there are resulting changes in survival. Indeed, adding 40mM L-arginine to *thyA⁻* strains, in both genetic backgrounds (MG1655 and MDS42), during thymidine starvation, substantially increased survival and this was accompanied by significant increases in pHi (Fig 4A–4C). Adding 40mM L-arginine to *thyA⁻* strains in high thymidine media did not result in major growth defects that could account for these dramatic increases in survival (Table 4 and S2 Fig).

Genetic deletions of the acetate metabolism genes *ackA* and *pta* have been shown to enhance *E. coli's* acid resistance [22,23]. Removal of the *ackA* gene and the subsequent decrease in acetate production and excretion result in less acetate permeating back across the cell membrane and delivering protons into the cytoplasm [55]. Deletion of *ackA* from the *ΔatpF*, *ΔrecO*, and *ΔspeB* strains results in significant alleviation of TLD (Fig 4D). The increased survival of each was accompanied by significant elevations in pHi during thymidine starvation (Fig 4E–4G). Deletion of *ackA* from the *ΔcydB* mutant resulted in no significant increase in survival (S11 Fig; see Discussion).

Part of the effects that we observe from deletion of *ackA* likely arise due to the decreased growth rate of *ΔackA* cells in our experimental conditions, but changes in growth rate do not fully explain the protection from TLD. We found that the *thyA* mutant grown at 30°C has a lower growth rate than either *ΔackA* or *ΔrecO ΔackA* at 37° (S2 Fig), but that its growth at 30° C had a much less substantial effect on survival than the additional mutations (S12 Fig). Also note that since the three double mutants with similar survival levels at 3h starvation have very different growth rates, changes in replication rates cannot account for all the increased survival associated with *ΔackA* cells (Table 4 and S2 Fig).

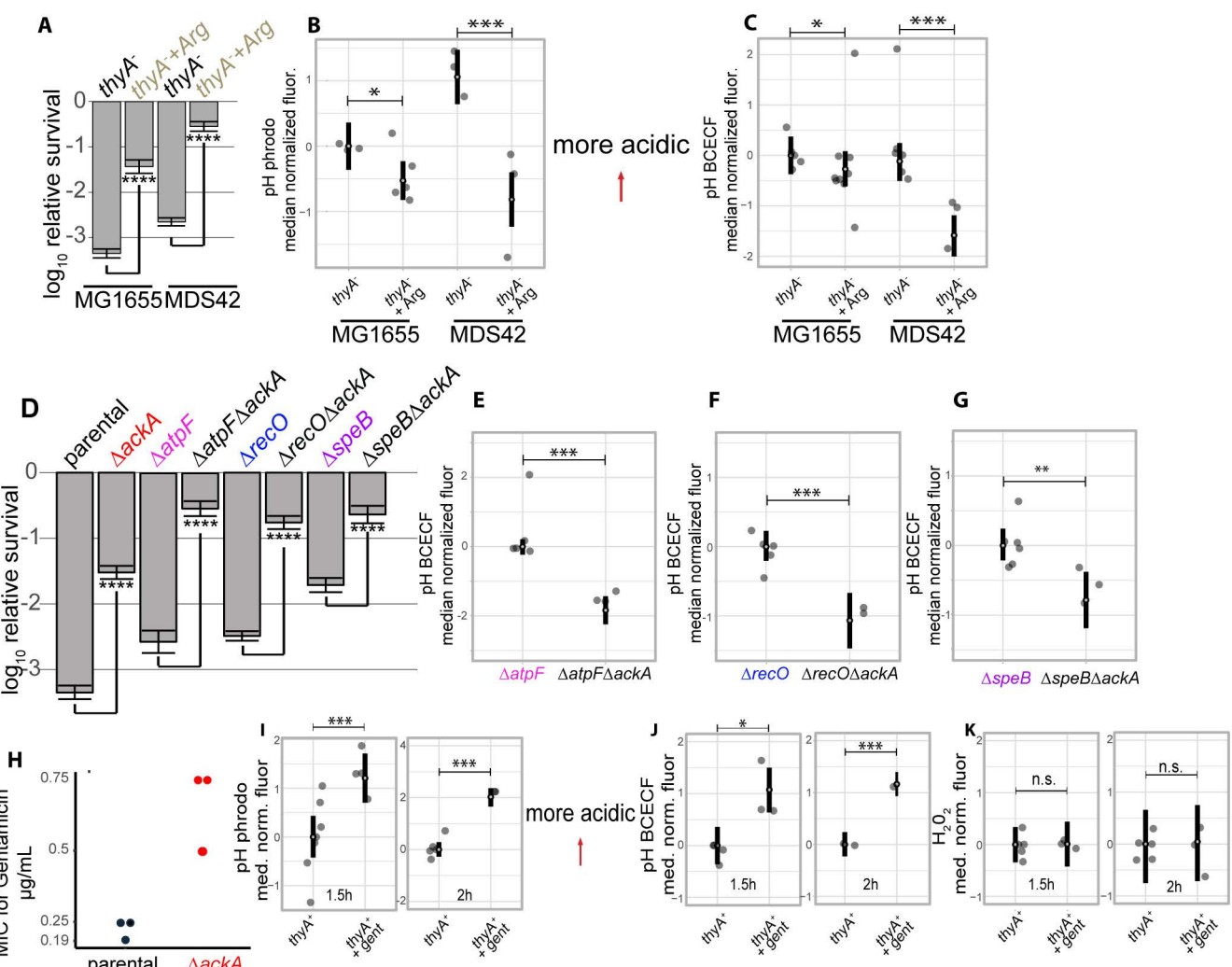

**Fig 4. Experimental manipulation of pHi modulates survival.** (A-C) Impact of arginine supplementation on survival and pHi. (A) L-arginine significantly increases the survival of *thyA*⁻ strains in the MG1655 and MDS42 backgrounds at 3h thymidine starvation. (B-C) Fluorescence was measured for each independent experiment for strains stained for pHi at 3h thymidine starvation and measured using flow cytometry using pHrodo Green (B) and BCECF-AM (C). (D-G) Impact of *ackA* deletion on survival and pHi. (D) Deletion of *ackA* from various resistant mutants results in significant increases in survival at 3h thymidine starvation. (E-G) Adjusted fluorescence of strains stained for pHi at 3h thymidine starvation and measured using flow cytometry using BCECF-AM. (H) Minimum inhibitory concentration (MIC) of the *thyA*⁻ and the Δ*ackA* strain for the antibiotic gentamicin. The concentrations are in μg/mL. (I-J) Adjusted fluorescence of wild type (*thyA*⁺) cells stained for pHi at 1.5h with and without 1μg/mL gentamicin treatment and measured using flow cytometry using pHrodo Green (I) or BCECF-AM (J). (K) Adjusted fluorescence of wild type (*thyA*⁺) cells stained for ROS accumulation at 1.5h with and without 1μg/mL gentamicin treatment and measured using flow cytometry using Peroxy Orange. The *p*-values in (A) and (D) were calculated as for the death assays in the rest of this work, using a Welch t-test. See Fig 3 caption for definitions of plotted values and significance tests for flow cytometry data.

## Intracellular acidification occurs following exposure to gentamicin

To test whether the genetic manipulation of pHi by deletion of *ackA* increases the resistance to other bactericidal perturbations, we measured the minimum inhibitory concentration (MIC) of the Δ*ackA* strain for various antibiotics. The MIC for gentamicin applied to the Δ*ackA* strain was three-fold higher than for the corresponding parental strain (Fig 4H). In order to test whether pH changes also accompany exposure to gentamicin, we stained *thyA*⁺ cells with pH-sensitive dyes, treated with a drug concentration that generated roughly 2 log₁₀-fold death

at 1.5h (S13 Fig), and looked for pH changes by flow cytometry. Both independent pH dyes showed significant acidification shifts 1.5h and 2h following drug exposure compared to the no-drug control (Fig 4I and 4J). We also stained these cells for ROS and observed no changes in ROS at 1.5h or 2h between the gentamicin treated cells and no-drug cells controls (Fig 4K). At higher concentrations of gentamicin, we saw both acidification and ROS changes (S14 Fig). These data suggest that the pH drop that is a key contributor to the killing process during thymidine starvation independently of ROS also occurs in other bactericidal perturbations, and may play a broader mechanistic role in cell death.

## Discussion

We used a set of complementary systems biology approaches to probe the underlying mechanisms of TLD in *E. coli*. We quantified the contribution of every non-essential gene in the genome to TLD via Tn-seq, evolved strains that exhibit unprecedented long-term TLD-resistance, and probed the transcriptional responses of TLD-sensitive and resistant strains. Dozens of novel candidate genes were identified across the various approaches. Among the recurring biological processes that modulated survival was intracellular pH homeostasis. Multiple independent lines of evidence support a critical role for cytoplasmic acidification in TLD. First, our Tn-seq results showed a significant enrichment of hits among genes that were shown, in a separate experiment [19] to have substantial impacts on survival under acid stress.

All three novel validated beneficial loss-of-function mutations arising from our study, *ΔspeB*, *ΔackA*, and *ΔatpF*, are in pathways that influence cytoplasmic pH. Mutations in the putrescine biosynthesis pathway, and in genes in or upstream of the operon encoding ATP synthase, recurred in the two best-surviving independently evolved isolates in different genetic backgrounds (MG1655 and MDS42). TLD sensitive strains showed intracellular acidification during thymidine starvation in flow cytometry experiments, even while controlling for changes in cell size, batch effects, and other potential artifacts. Survival of all of the tested strains strongly associates with intracellular acidification throughout the death process. In addition, when we stained a starved *thyA*- population with pH sensitive dyes and sorted into low and high fluorescence at 1.5h, we found that cells showing lower acidification survived significantly better than those showing higher acidification. Finally, manipulations that increased the pHi resulted in corresponding increases in survival during thymidine starvation—in some instances without significant changes in ROS accumulation.

Previous researchers have proposed, and our data supports, a model in which replication initiation during thymidine starvation causes DNA damage. A subset of SOS-induced genes are significantly upregulated in the TLD-resistant strain and show a survival cost when disrupted in the *thyA*- strain, confirming that some DNA repair genes have protective roles during thymidine starvation. Other DNA damage repair proteins, such as *recO*, aggravate the damage. The enhanced survival of *ΔrecO* cells is associated with both lower ROS levels and less intracellular acidification. Although *ΔrecO's* involvement in TLD is not new, our finding that the knockout lowers both ROS levels and intracellular acidification suggests a potential link between DNA damage and ensuing acidification and ROS generation.

The involvement of ROS in cell death during TLD is controversial [56], and it is linked to the contentious hypothesis that ROS is involved in a unifying killing mechanism shared by several different classes of bactericidal antibiotics. The role of ROS in antibiotic lethality has been challenged in studies that showed that antibiotics can kill in the absence of ROS [57,58]. Our work presented here also shows thymineless death in the absence of ROS. A newer model that seeks to resolve these paradoxes suggests that killing can derive either from a primary damage or from a secondary stress response mediated by ROS [59,60].

We found that the disruption of several genes involved in producing ROS (*cydABX*, *nadAB*, *ubiC)* increased survival during T-starvation; the transcription of these same genes was down-regulated in the TLD-resistant mutants compared to sensitive strains (Fig 2H). While our survival profiling experiment confirmed previous findings that catalase absence has no effect on TLD [56] (S1 Table), we also found that disruptions in *gstA*, a gene encoding a hydrogen peroxide detoxifying enzyme, hurt survival and its transcription was upregulated in the TLD-resistant evolved strain compared to sensitive strains (Fig 2I).

Our flow cytometry experiments, in which both ROS and cytoplasmic acidification signals were carefully normalized by cell volume, revealed that while both ROS and intracellular acidification correlated with TLD survival at 3h T-starvation, the *thyA*- strain showed no ROS accumulation at 1.5H (a time point in which we did see significant intracellular acidification and in which we already begin to see killing). We also see cytoplasmic acidification in wildtype unstarved cells treated with antibiotic and no changes in ROS at concentrations that cause death (S13 Fig). We saw both cytoplasmic acidification and ROS accumulation at higher concentrations of the drug. Whether intracellular acidification is the primary damage or an earlier lethal stress response than ROS remains unclear.

We propose that thymidine starvation results in three toxic outcomes in the cell: DNA damage, intracellular acidification, and ROS accumulation. We observe acidification before evidence of ROS accumulation. It is possible that ROS and acidification are two independent responses and acidification simply occurs earlier. It is also possible that acidification triggers ROS accumulation. The fact that targeted gene deletions that we identified in pHi-altering pathways also alleviate ROS accumulation during thymidine starvation (and especially that in some cases beneficial deletions affect TLD without bringing down ROS levels; Fig 3H–3M) argues for the latter interpretation.

Several groups have previously suggested that acid and oxidative stress can synergize. Many acid stress genes overlap with those of oxidative stress, and researchers have proposed that low pH amplifies the toxicity of radicals [61–63]. Glutamate and arginine have also been reported to play important roles in the protection against oxidative stress under acidic conditions in *E. coli* [64], partly through enhanced nitric oxide production, and thus contributions orthogonal to effects on pH cannot be ruled out. Oxidation reactions are affected by pH, so an increase in pH achieved via these acid resistance systems could modulate the outcome of oxidative reactions and processes [65,66].

Both intracellular acidification and ROS can damage DNA. Deletion of *ackA* has been reported to have protective effects on DNA integrity [67]. Previous groups have shown that *ackA* and *pta* are associated with DNA replication and/or repair through as yet unknown mechanisms [67,68]. A 2011 screen of the *E. coli* Keio collection under ~300 conditions, found *ackA* to be the top surviving strain under high azidothymidine [69]. Azidothymidine is a thymidine analogue that arrests replication and produces single-strand DNA gaps in *E. coli* [70]. It is possible that the disruption in acetate production caused by loss of *ackA* somehow affects DNA replication and/or repair through a reduction in acidification. It is also possible that ROS accumulation and acidification contribute to TLD in ways separate from their causation of DNA damage.

The lack of increased survival in Δ*cydB*Δ*ackA*, relative to either mutant in isolation, is potentially informative. It has been reported that Δ*cydB* upregulates several acid resistance genes and exhibits acid resistance [71], and may explain the lack of acidification that we observed during thymidine starvation in this mutant. This group also reported that in Δ*cydB* *c*ells, *poxB* is upregulated [71]. *poxB* encodes pyruvate oxidase, the key enzyme in an alternate, low-efficiency pathway to synthesize acetate [72]. There have been two additional reports of increased activity of the PoxB pathway in a Δ*cydB*Δ*cyoB*Δ*nuoB* triple mutant [73,74]. If *ackA*

expression is already low in the Δ*cydB* strain in favor of *poxB* expression, removing *ackA* gene from the Δ*cydB* strain may result in minimal additional benefits. A potential role for the PoxB enzyme in alleviation of acidification and/or ROS accumulation during TLD is an interesting avenue for future research due to its ability to transfer electrons directly to the terminal oxidases and potentially uncouple respiration from ATP synthesis (and from the coincident proton influx) [75–77].

Several questions remain, particularly about the nature of the path that leads from thymidine starvation to acidification and (later) ROS accumulation. A recent study reported that trimethoprim treatment induces an acid stress response in *E. coli*; the researchers proposed that adenine depletion may cause a drop in pH [78]. Another group more recently linked antibiotic-induced adenine limitation to oxidative stress [79]. Thymidine starvation, DNA damage, and attempts to repair the damage, all cause nucleotide pool disruptions [80–83]. It is possible that dysregulation in adenine levels is the source of both the pH drop and the resulting increase in ROS levels during thymidine starvation. ATP levels may play an additional role in TLD independent of ROS and pH effects, and may help explain the enhanced survival of the Δ*atpF* strain.

Another outstanding question is whether TLD is an active or passive process. It is interesting to note that the transcription of siderophores involved in iron acquisition and ROS generation are significantly upregulated in the *thyA⁻* strain in the comparison of starved versus unstarved conditions and downregulated in the evolved starved versus unstarved conditions. This suggests possible triggering of an active ROS death pathway. Alternatively, these processes may be activated in response to the pH drop in the TLD-sensitive strain. Previous work has shown that lowering of pH increases the availability of free iron [84].

A more parsimonious model is that TLD is a passive process and that ROS, acidification, and DNA damage constitute a perfect storm. Under normal conditions, drops in pH or spikes in ROS levels would cause only minor damage to DNA that the cell can recover from. However, when this damage is compounded by previous DNA damage caused by replication initiation and repair during thymidine starvation, the results are lethal. Regardless of whether TLD is a passive or active process, we conclude that intracellular acidification plays an instrumental role in the sequence of events from thymidine starvation to cell death. Our findings that genetic manipulations that increase pH also increase gentamicin resistance, and that exposure to gentamicin results in significant pH drops provide intriguing evidence that intracellular acidification may be a common causal factor in bacterial cell death caused by other bactericidal perturbations.

## Methods

### Strains and culture

Thymidine auxotrophs were generated in two genomic backgrounds, MG1655, the prototype K-12 strain, and the "Multiple Deletion Series 42" (MDS42), a reduced-genome derivative of MG1655 [25] (See Table 5 for full list of strains used). The latter strain has 704 nonessential genes as well as insertion sequence elements and cryptic prophages deleted, amounting to a 14% reduction in the genome.

The *thyA* gene has a transcription terminator within the coding sequence that acts as the stop signal for the upstream essential gene *umpA* [87–89]. In order to remove t*hyA* function and preserve viability, we made an internal deletion without disrupting the overlapping transcript [88] via one-step inactivation using PCR products [90]. The *thyA* mutation in both starting strains is identical and has been verified with PCR, sequencing, and by failure to grow in thymidine-free media and agar plates. Both starting strains exhibit rapid and steep death in thymidine-free media.

**Table 5. Strains used in this work.**

| Strain | Genotype | Source |
|---|---|---|
| *thyA*+ | MG1655 | MG1655 ATCC 700926 [85] |
| MG1655 *thyA*- | MG1655 *thyA*- | this work |
| MDS42 *thyA*+ | MDS42 | [86] via Alison Hottes |
| MDS42 *thyA*- | MDS42 *thyA*- | this work |
| Δ*ackA* | MG1655 *thyA*- Δ*ackA* | this work |
| Δ*cydB* | MG1655 *thyA*- Δ*cydB* | this work |
| Evolved MG1655 | *thyA*- see S6 Table for list of mutations | this work |
| Evolved MDS42 | *thyA*- see Table 3 for list of mutations | this work |
| Sibling isolate of Evolved MDS42 | MDS42 *thyA*- all mutations in Table 3 except *atpF* is wild type | this work |
| Δ*atpF* | MG1655 *thyA*- Δ*atpF* | this work |
| Δ*recO* | MG1655 *thyA*- Δ*recO* | this work |
| Δ*atpF*Δ*recO* | MG1655 *thyA*- Δ*atpF* Δ*recO* | this work |
| Δ*speB* | MG1655 *thyA*- Δ*speB* | this work |
| Δ*speB*Δ*recO* | MG1655 *thyA*-Δ*speB* Δ*recO* | this work |
| Δ*atpF*Δ*ackA* | MG1655 *thyA*- Δ*atpF* Δ*ackA* | this work |
| Δ*recO*Δ*ackA* | MG1655 *thyA*- Δ*recO* Δ*ackA* | this work |
| Δ*speB*Δ*ackA* | MG1655 *thyA*- Δ*speB* Δ*ackA* | this work |
| Δ*cydB*Δ*ackA* | MG1655 *thyA*- Δ*cydB* Δ*ackA* | this work |

Single gene deletions were obtained from the Keio collection [91] and transferred to the *thyA*- strain in the MG1655 background by P1 transduction [92] followed by selection on LB/kanamycin/thymidine plates. Kanamycin-resistant clones were tested for the clean deletion by PCR, and then cured of the resistance cassette by transformation with the plasmid pcp20 [93] prior to characterization.

All strains in the *thyA*- background, unless they were being starved, were supplemented with 50μg/ml of thymidine (Sigma Aldrich, T1895) in liquid media (called "high thymidine" in this work) and all plates contained 15g/L agar and 50μg/ml of thymidine, unless otherwise stated.

For all experiments in liquid, strains were grown in defined supplemented M9 media either at 30°C or 37°C. In all experiments, unless otherwise stated, the liquid media was the same: 1x M9 salts (BD 248510) supplemented with a synthetic complete amino acid supplement (US Biological, D9515) that also supplies uracil and adenine; cytosine and guanine (each at 133μM); a vitamin solution (each in the 10μM-20μM range) of thiamine, folic, riboflavin and alpha lipoic acid; micronutrients: boric acid, cobalt chloride, copper sulfate, manganese chloride, zinc sulfate, ammonium molybdate, ferric citrate, MgS04, CaCl2 (each in the 10μM-10mM range, following the same formulation as in MOPS media [94]); and 0.2% glucose. Growth rates in high thymidine in our rich defined media were higher than in LB for every strain we tested (Table 6).

## Thymineless death assays

Overnight cultures of bacteria grown from single colonies were diluted 1:200 into 2mL high thymidine media and grown for 2h with shaking at 37°C. They were washed twice and placed in thymidine-free media, plated at 0h on LB/thymidine plates and again after the stated time of thymidine starvation with shaking at 37°C. The number of independent experiments for 3h

**Table 6. Growth rates of strains in rich defined media vs. LB.**

| Strain | Average doublings per hour, 37°C |
|---|---|
| MG1655 *thyA*⁺ | 2.09 |
| MG1655 *thyA*⁺ in LB | 1.97 |
| MG1655 *thyA*⁺+Arg in zero thy | 1.96 |
| MG1655 *thyA*⁺+Arg in zero thy in LB | 1.76 |
| MG1655 *thyA*⁻ | 2.26 |
| MG1655 *thyA*⁻ in LB | 1.67 |
| MG1655 *thyA*⁻ + Arg | 2.07 |
| MG1655 *thyA*⁻ + Arg in LB | 1.94 |
| Evolved MG1655 | 1.82 |
| Evolved MG1655 in LB | 1.36 |
| Parental MDS42 | 2.29 |
| Parental MDS42 in LB | 1.52 |
| Evolved MDS42 | 1.51 |
| Evolved MDS42 in LB | 1.30 |

Death assays: *thyA*⁻ in MG1655: n = 18; Δ*ackA*: n = 15; Δ*cydB*: n = 9; evolved MG1655: n = 7; *thyA*⁻ in MDS42: n = 9; evolved in MDS42: n = 8; Δ*atpF*: n = 8; Δ*recO*: n = 7; Δ*atpF*Δ*recO*: n = 7; Δ*speB*: n = 11; Δ*speB*Δ*recO*: n = 7; Δ*atpF*Δ*ackA*: n = 4; Δ*recO*Δ*ackA*: n = 6; Δ*speB*Δ*ackA*: n = 6. The number of independent experiments for the long-term death assay (Fig 2C): *thyA*⁻ MG1655: (3h) n = 18, (4h) n = 3, (24h) n = 1; evolved in MG1655: (3h) n = 7, (22h) n = 2, (24h) n = 3; *thyA*⁻ MDS42: (3h) n = 9, (24h) n = 3; evolved MDS4: (3h) n = 8, (22h) n = 3, (24h) n = 3, (48h) n = 3, (72h) n = 2. The number of independent experiments for the 22h death assay (Fig 2D): *thyA*⁻ in MDS42: n = 3; *atpF* wild type (sibling isolate) strain: n = 3; *atpF* early stop (evolved in MDS42): n = 3.

**Death assays with L-arginine supplementation.** Overnight cultures were prepared as above. After 2 washes, cells were placed in thymidine-free media along with 40mM L-arginine, and plated for colony counting on LB/thymidine plates at 0h and again after 3h of thymidine starvation (with arginine) shaking at 37°C. For the arginine supplemented death assays: *thyA*⁻ in MG1655: n = 6; *thyA*⁻ in MG1655 + Arg: n = 4; *thyA*⁻ in MDS42: n = 3; *thyA*⁻ in MDS42 + Arg: n = 3; Δ*ackA*: n = 15; Δ*ackA* + Arg: n = 5.

**Death assays with antibiotic treatment.** After the pregrowth and washes as outlined above, thyA⁺ cells were treated with 1μg/mL gentamicin and plated for colony counting on LB/thymidine plates at 0h and again after 1.5h of shaking in defined media at 37°C. The number of independent experiments for this death assay at 1.5h: thyA⁺: n = 4; thyA⁺+ gent1: n = 5.

## Characterization of growth rates

Overnight cultures of bacteria grown from single colonies were diluted 1:200 in 150μl of fresh media and grown in a Biotek Synergy MX plate reader shaking continuously for up to 24h at 37°C. The absorbance at 600nm (OD 600) was measured every 10 minutes. Doubling times were calculated and provided in Table 4.

## Survival profiling

**Transposon insertion library generation.** Using Gibson assembly, we redesigned the EZ-Tn5 Transposon Vector (Epicentre) so that the transposon insertion has, in addition to the kanamycin cassette, an Illumina sequencing adapter. MG1655 *thyA*⁻ cells were prepared

for transformation and electroporated with 2μL of the generated transposomes. After 1h, serial dilutions were made of a small aliquot onto LB/kanamycin/thymidine plates in order to gauge the number of successful transformants. The remainder of the cell mixture was added to 250mL of LB/kanamycin/thymidine and grown to amplify the mutants. After 8h, the library was pelleted and resuspended in a small volume of 15% glycerol (diluted in LB), subsequently aliquoted, snap-frozen, and stored at -80°C. Our library had 174,101 insertions in the genome, or ~1 insertion every 27 base pairs.

**Selection of transposon library.** The library was thawed and pre-grown for 2h in high thymidine, washed twice in 1x phosphate buffered saline (PBS), and placed in thymidine-free media. Once the library was placed in thymidine-free media, a "Start" sample was collected for DNA extraction. Another sample, "0h", was taken at the same time but placed in high thymidine and allowed to undergo 2–3 doublings before DNA extraction. After 3h of thymidine starvation, a final sample, "3h" was placed in high thymidine for an outgrowth of 2–3 doublings before DNA extraction.

To summarize, the "Start" sample was taken at the very beginning of the thymidine starvation selection and was never placed in high thymidine for an outgrowth. Conversely, both "0h" and "3h" samples were thymidine-starved for the designated amount of time (0h and 3h) and an aliquot was placed in high thymidine for an outgrowth of 2–3 doublings before DNA extraction. The doublings were measured by cell count on dry runs prior to the experiment. Footprinting after the particular selection applied here requires an outgrowth period, because after 3h thymidine starvation, the insertion library contained a mixed population of dead cells and survivors. Without the outgrowth with high thymidine there would be no way to identify which cells were alive after the selection (the footprinting would pick up the signal from both live and dead cells and both would be sequenced and mapped). Therefore, the outgrowths serve to amplify signal from live cells relative to any residual DNA from dead cells. However, in an outgrowth setup like the one we deployed there is a danger in selecting for survivors that recover more rapidly. The outgrowth at 0h serves as a control for this process: in order to control for any rapid growers during the high thymidine outgrowth, our survival scores are always calculated by normalizing the outgrowth after 3h thymidine starvation by the identical outgrowth performed at the 0h starvation time point (*i.e.*, the 0h and 3h outgrowths do not indicate different periods of outgrowth, but rather, indicate the duration of thymidine starvation prior to beginning outgrowth). The "Start" aliquot allowed us to identify insertion mutants with significant growth defects from the initial library generation.

**Footprinting and sequencing.** Genomic DNA was isolated from the samples collected during the selection using the DNEasy Blood and Tissue Kit (Qiagen) and purified using Zymo genomic DNA clean & concentrate kit (Zymo Research). The isolated and purified DNA was digested with MspI and HinP1 in separate digests that are subsequently pooled. After purification again using a Zymo clean and concentrate kit, a Y-linker [95] was annealed and ligated to the DNA and the samples were purified using Agencourt AMPure beads (Beckman Coulter). The Y-linker was annealed by combining the primers with annealing buffer in a PCR tube at 90°C for 2 min, followed by cooling to 30°C at a rate of ~2 degrees per minute. After snap cooling, the Y-linker was promptly used or frozen for future use. A second Illumina bar code was added by PCR (the first was added to the transposon fragment itself before library generation). After another bead cleanup was a second PCR amplification, which added dual indexes for sequencing. After a final bead cleanup, samples were pooled and sequenced using a NextSeq 500 sequencer (Illumina).

Y-linker primers:
ACTACGCACGCGACGAGACGTAGCGTC

5'phos—CGGACGCTACGTCCGTGTTGTCGGTCCTG
Primers to add the second Illumina adaptor:
ACACCTAACCGCTAGCACGTAATACGACTC
GTGACTGGAGTTCAGACGTGTGCTCTTCCGATCTACTACGCACGCGACGAGACG

**Survival profiling analysis.** The bcl2fastq package from Illumina was used to separate the sequencing reads based on sample number and to obtain FASTQ files for each sample. Preprocessing & alignment steps clipped the Illumina adaptors & transposon sequence to identify the target sequence [96]. Quality score filtering dropped reads if the target was less than 15bps and trimmed low quality ends of reads [95]. Bowtie aligned reads to a given genome database and the total reads aligned per sample were quantified. A python script enumerated the number of footprints present within each gene, and generated data tables with the set of counts. Reads were normalized to fragments per million (FPMs). The R package rateratio.test was used to calculate significance for the differences in insertion frequencies for each gene at the 3h vs. 0h time points, aggregated at the gene level. A survival score for each gene was defined as the $\log_2$ fold change ($L_2$FC) of FPM at 3h/ FPM at 0h and p-values calculated assessing the significance of the observed count differences via the exact rate ratio test [97]. Two-sided exact tests and matching confidence intervals for discrete data. R Journal, 2(1), 53–58.], and subsequently corrected for multiple hypothesis testing using the Benjamini-Hochberg method.

Candidate lists were generated by collecting all genes with a significant (FDR<0.05) survival score over a threshold $\log_2$ fold change magnitude (+/- 0.5). For each of these genes, if its survival score calculated for FPM at 3h / FPM at "Start" had an opposite sign compared to the survival score of FPM at 3h/ FPM at 0h, they were discarded from the candidate list.

**Secondary GO term enrichment analysis.** For each of the conditions noted in the text (benzoic acid exposure, nalidixic acid exposure, sodium chlorite exposure), we utilized data for *E. coli* BW25113 exposed to the given stress, taken from the Fitness Browser Database [19] and partitioned as follows: Each gene was assigned as being either beneficial in that condition (fitness score < -1 and t-score < -2), deleterious in that condition (fitness score > +1 and t-score > +2), or neutral (neither of the preceding conditions applied). In the case of the benzoic acid deleterious genes only, we used a fitness score threshold of +0.5 instead of +1 due to the relatively weak fitness scores observed in that case. The 'deleterious' genes were excluded from further analysis, and then we considered the remaining set of beneficial genes as those associated with the process of interest, compared with the 'neutral' gene for that set as a background distribution.

## Laboratory evolution

Four replicates of each starting strain (four *thyA⁻* in MG1655 and four *thyA⁻* in MDS42) were selected for survival and growth in media supplemented with 0.2μg/mL thymidine as opposed to the typically required level of 20μg/mL. The selections began by making a 100,000-fold dilution from saturated cultures of the starting strains grown overnight in media supplemented with 50μg/mL thymidine. The overnight cultures, grown to saturation before the initial inoculation, were at 37°C but once the selections began, the cells were kept aerated at 30°C to slow the rapid death process and boost the number of survivors at each transfer.

The selection process lasted for ~50 daily transfers. All but the last 7 daily transfers involved a 2-fold dilution in which the 2mL of 24h old cells and media were vortexed, 1ml was removed, pelleted, and reconstituted in 2ml fresh media. For the last 7 transfers, the dilution

was increased because the number of survivors had increased; these transfers involved a 200-fold dilution of 24h cultures directly into 2mL fresh media (i.e. no pelleting).

Throughout the selection process, weekly or biweekly, the evolved batch cultures were plated onto LB/thymidine plates both at 0h and 24h post transfer in order to keep track of any large changes in survival numbers. In addition, weekly or biweekly, 15% of the 24h batch cultures were mixed with LB and glycerol and frozen at -80°C.

After the 50 transfers, the four heterogeneous evolved populations in each background were characterized for enhanced survival and the best surviving from each background were spread onto LB/thymidine plates in order to isolate single colonies. These isolates were re-verified by PCR with primers flanking the *thyA* gene and, in the case of the strains in the MDS42 background, using primers flanking genes deleted in this background. Six colonies from the top surviving evolved population in the MG1655 background and six from the top surviving evolved population in the MDS42 background, once verified, were grown overnight in LB+50μg/mL thymidine and frozen in LB and glycerol at -80°C. These twelve isolates were characterized for increased survival in thymidine-free media. The two top surviving isolates—one from each background—(assessed via half-life in thymidine-free media) are discussed in the main text. We also discuss a sibling of the top surviving isolate in the MDS42 background which shared all the mutations of the main characterized MDS42 isolate except for the early stop mutation in *atpF*.

## Whole genome sequencing

DNA was isolated using the DNeasy Blood and Tissue kit (Qiagen), then barcoded and prepared for sequencing using the Nextera XT DNA Library Prep Kit (Illumina). All samples were pooled and sequenced using a NextSeq 500 sequencer (Illumina). After the sequencing reads were demultiplexed and preprocessed, as described above for the survival profiling sequencing data, the mutations present in the evolved strains compared to the parental background were identified using breseq (version 0.23) [98].

## RNA sequencing

The transcriptome profiling experiment mapped and quantified RNA extracted from three experimental conditions in the evolved and parental strains in the MDS42 background. Both strains were diluted 1:200 from overnight cultures into high thymidine and grown for 2h at 30°C before strains were washed twice and placed in new media. Three samples were taken for each strain: a high thymidine sample 2h into growth, a sample 30 minutes into thymidine starvation, and a control was washed twice similar to the starved sample but placed back into high thymidine and collected after 30 minutes. When the evolved strain in high thymidine was compared to the parental in high thymidine, the samples before washes and pelleting were used. When the evolved in the starved condition was compared to the evolved unstarved, the high thymidine sample post washes and pelleting was used.

After incubation with RNAprotect Bacteria Reagent (Qiagen), samples were stored at -80°C. Total RNA was made from the pellets using the Total RNA Purification Plus Kit (Norgen) according to the manufacturer's recommended protocol for Gram-negative bacteria. Ribosomal RNA was removed from these samples using the Ribo-Zero rRNA Removal Kit (Illumina). Indexed libraries were prepared from the resulting RNA using the NEBNext Ultra Directional RNA Library Prep Kit for Illumina (New England Biolabs). Samples were pooled and sequenced on a NextSeq 500 sequencer (Illumina).

The sequencing reads were demultiplexed, and preprocessed using cutadapt and trimmomatic, as described above for the survival profiling data, and then Rockhopper (version 2.03)

[99,100] was used in two modes. First Rockhopper was used on each RNA read by itself to get quantitation and statistics on expression levels, and then it was used in paired mode on specified pairs of conditions in order to obtain significance statistics for changes in expression levels. In paired mode Rockhopper uses a negative binomial distribution to estimate the uncertainties in the read counts. The *p*-values are FDR-corrected using the Benjamini-Hochberg procedure [101]. The counts for each gene from the Rockhopper output were expressed in the normalized form, RPKM. Transcript counts were also normalized to transcripts per million (TPMs) in order to allow for more direct comparison with the survival profiling data.

## Genes with concordant effects across experimental approaches

Gene disruptions showing a significant survival score above 0.5 and a $\log_2$ fold change of RNA TPMs below -0.5 in any of the three comparisons (the evolved vs. parental in starved; evolved vs. parental in unstarved; and evolved starved vs. evolved unstarved) were compiled into three lists of genes that putatively exacerbate survival during thymidine starvation. Gene disruptions showing a significant survival score below -0.5 and a LFC of TPMs above 0.5 in any of the three comparisons detailed above were compiled into 3 lists of genes that putatively alleviate survival during thymidine starvation.

## Flow cytometry

Fluorescence intensity was determined by fluorescence-based flow cytometry, using a BD LSR-FORTESSA instrument at the Columbia University Herbert Irving Comprehensive Cancer Center Flow Cytometry Core. Bacterial cells were grown for 2h in high thymidine at 37°C after 1:400 dilution from overnight cultures. After this pregrowth, cells were washed twice and placed into thymidine-free media (unless otherwise stated), with shaking at 37°C. Cells were stained as detailed below for the various dyes. No-dye cells were included to control for autofluorescence. Fluorescence was calculated at 1.5h and 3h into thymidine starvation. A total of ~50,000 events for each time-point sample were determined. Gating was used to exclude cellular debris. Data were extracted using FCS Express 6.

After data acquisition, the autofluorescence was subtracted by using the data from the no-dye cells (pooled across all genotypes/conditions in a particular analysis batch) to fit a LOESS model predicting the fluorescent signal of interest as a function of the forward scatter (FSC) and side scatter (SSC) signals. The LOESS-predicted autofluorescence was then subtracted from the signal observed for each cell from the experimental samples. After the LOESS nodye-subtraction, the fluorescence signals were fit to a mixed effects model (using the lme4 R package) of the form:

$\log(\text{fluorescence}) \sim a \log_2(\text{fsc}) + b \log_2(\text{ssc}) + c\_\text{strain} + \text{batch\_effect} + \text{rep\_effect}$.

Here a and b are fixed effects indicating how the signal scales, on average, with FSC and SSC (thus explicitly accounting for changes in cell size/shape using the FSC/SSC as proxies, by factoring out any systematic changes in fluorescence that can be explained by changes in these parameters), and c_strain is a strain/condition-specific average fluorescence after accounting for the FSC/SSC effect. Batch_effect and rep_effect are random effects representing the biological replicate and day when the sample data were taken, and the inclusion of these terms allows us to factor out any systematic effects arising from day to day variations in assay behavior. The c_strain parameter was our key parameter of biological interest, and we calculated profile-based 95% confidence intervals for these parameters. In addition, the significance of differences in this parameter between biological conditions of interest was assessed using the linearHypothesis function of the R car package, against a null hypothesis that two groups had the same value of the c parameter. Only cells with both $\log_2(\text{fsc})$ and $\log_2(\text{ssc})$ greater than 8

were included in the analysis, as we found on visual inspection that events with lower values were likely simply noise.

In addition, for plotting purposes, we obtained corrected cell-level fluorescence data by calculating, for each cell, the quantity

$log\_fluor\_corrected = \log_2(fluorescence) - a * \log_2\_fsc - b * \log_2\_ssc$

Where $\log_2\_fsc$ and $\log_2\_ssc$ refer to the specific $\log_2(fsc)$ and $\log_2(ssc)$ values measured for that cell. Taken together, the model yields a corrected set of fluorescence data after accounting for cell size/shape in a consistent way, as well as estimates (with confidence intervals) of the average fluorescence for each strain after correcting for size effects.

**BCECF-AM.** (Cat no. B1150, Thermo Fisher): After pregrowth, washes, and thymidine-starvation, as detailed above, cells were incubated with 20μM BCECF in thymidine-free media (unless otherwise stated) for the last 30 minutes of the designated time period followed directly by visualization (i.e. for a 3h starvation before visualization, dye was added at 2.5h). Two channels were selected for acquiring paired sets of images: a pH-sensitive wavelength using a 488 nm laser (band pass filter 530/30) and pH-insensitive wavelength using 405 nm laser (band pass filter 525/50). In the main text and supplement, the pH-sensitive MFIs are shown.

The number of independent experiments for visualization at 1.5h: *thyA*⁺: n = 6; *thyA*⁻ unstarved: n = 3; *thyA*⁻ starved: n = 5; *ΔatpF*: n = 5; *ΔrecO*: n = 5; *ΔspeB*: n = 6; *ΔackA*: n = 4; *ΔcydB*: n = 5. The number of independent experiments for visualization at 3h: no dye control: 9 different strains were tested at 2–3 time points and none showed autofluorescence; *thyA*⁺: n = 6; *thyA*⁻ unstarved: n = 8; *thyA*⁻ starved: n = 7; evolved MG1655: n = 4; *thyA*⁻ MDS42 starved: n = 4; evolved MDS42: n = 4; *ΔatpF*: n = 6; *ΔrecO*: n = 6; *ΔspeB*: n = 6; *ΔackA*: n = 5; *ΔcydB*: n = 7; *ΔrecOΔackA*: n = 3; *ΔspeBΔackA*: n = 3; *ΔatpFΔackA*: n = 3.

*Flow cytometry for arginine supplementation assay using BCECF*: In the cases of L-arginine supplementation (and its controls) cells were incubated with 40mM L-arginine for the first 2.5h of thymidine starvation, washed twice to remove the amino acid, and incubated with dye for 30 minutes in thymidine-free media before visualization. The no-arginine control was washed and stained exactly like the cells with added arginine. The number of independent experiments for visualization at 3h: *thyA*⁻ MG1655: n = 5; *thyA*⁻ MG1655 + Arg: n = 7; *thyA*⁻ MDS42: n = 5; *thyA*⁻ MDS42 + Arg: n = 3.

*Flow cytometry for gentamicin assay for WT cells stained with BCECF*: After the pregrowth and washes as outlined above, *thyA*⁺ cells were treated with 1μg/mL gentamicin. After 1h (30 minutes before visualization) cells were stained with BCECF dye as outlined above. At 1.5h treatment, acidification was visualized using flow cytometry. The no drug cells were prepared, stained and visualized exactly as the drug-treated cells except for the addition of gentamicin. The number of independent experiments for visualization at 1.5h: *thyA*⁺: n = 6; *thyA*⁺+ gent(low): n = 3.

**pHrodo Green AM pHi Indicator.** (Cat no. P35373, Thermo Fisher): 10μL of the pHrodo dye was added to 100μL of the Powerload concentrate (as specified in the pHrodo Green kit). 3.3μL of this combination was added to 300μL of cells (after pregrowth, washes, and thymidine-starvation—unless otherwise stated—as detailed above) and incubated for the last 30 minutes of the designated period at 37°C followed directly by visualization. Data were gathered at the stated times points using the following detection parameters: 488 laser, with 530/30 nm band pass filter. The number of independent experiments for visualization at 1.5h: *thyA*⁺: n = 4; *thyA*⁻ unstarved: n = 3; *thyA*⁻ starved: n = 8; *ΔatpF*: n = 4; *ΔrecO*: n = 3; *ΔspeB*: n = 6; *ΔackA*: n = 3; *ΔcydB*: n = 3. The number of independent experiments for visualization at 3h: no dye control: n = 3; *thyA*⁺: n = 3; *thyA*⁻ unstarved: n = 6; *thyA*⁻ starved: n = 12; evolved MG1655: n = 8; *thyA*⁻ MDS42 starved: n = 8; evolved MDS42: n = 3; *ΔatpF*: n = 8; *ΔrecO*: n = 8; *ΔspeB*: n = 6; *ΔackA*: n = 7; *ΔcydB*: n = 7.

*Flow cytometry for arginine supplementation assay using pHrodo Green*: In the case of exogenous L-arginine addition (and its controls), cells were incubated with 40mM L-arginine for the first 2.5h of thymidine starvation, washed twice to remove the L-arginine, and the cells were incubated with dye for 30 minutes in thymidine-free media before visualization. The no-arginine control was washed and stained exactly like the cells with added arginine. The number of independent experiments for visualization at 3h: *thyA*$^-$ MG1655: n = 4; *thyA*$^-$ MG1655 + Arg: n = 6; *thyA*$^-$ MDS42: n = 3; *thyA*$^-$ MDS42 + Arg: n = 3.

*Flow cytometry for Gentamicin assay for WT cells stained with pHrodo Green*: After the pregrowth and washes as outlined above, *thyA*$^+$ cells were treated with 1μg/mL gentamicin for the lower dose ([Fig 4I]) or with 4μg/mL gentamicin for the higher dose ([S14 Fig]). 30 minutes before visualization cells were stained with pHrodo Green as outlined above. At 1.5h treatment, acidification was visualized using flow cytometry. The no drug cells were prepared, stained and visualized exactly as the drug-treated cells aside from the addition of gentamicin. The number of independent experiments for visualization at 1.5h: *thyA*$^+$: n = 5; *thyA*$^+$+ gent(low): n = 3; *thyA*$^+$+ gent(high): n = 2.

**Peroxy Orange 1, Tocris.** (Cat no, 49-441-0, Fisher Scientific): After pregrowth and washes as detailed above, cells were placed into thymidine-free media along with 5μM dye and incubated, shaking, at 37°C, for the designated amount of time before visualization. Data were gathered at the stated times points using the following detection parameters: 561 nm laser, 585/40 nm band-pass filter. The number of independent experiments for visualization at 1.5h: *thyA*$^+$: n = 3; *thyA*$^-$ unstarved: n = 4; *thyA*$^-$ starved: n = 3; *ΔatpF*: n = 3; *ΔrecO*: n = 3; *ΔspeB*: n = 3; *ΔackA*: n = 3; *ΔcydB*: n = 3. The number of independent experiments for visualization at 3h: no dye control: n = 3; *thyA*$^+$: n = 6; *thyA*$^-$ unstarved: n = 6; *thyA*$^-$ starved: n = 10; evolved MG1655: n = 3; *thyA*$^-$ MDS42 starved: n = 10; evolved MDS42: n = 3; *ΔatpF*: n = 16; *ΔrecO*: n = 5; *ΔspeB*: n = 12; *ΔackA*: n = 14; *ΔcydB*: n = 9; *ΔrecOΔackA*: n = 5.

*Flow cytometry for exogenous H$_2$0$_2$ control*: at 1h thymidine starvation, 1mM hydrogen peroxide was added to cells stained with Peroxy Orange dye as detailed above and incubated for 30 minutes at 37°C before visualization. The number of independent experiments for visualization at 1.5h: *thyA*$^-$ unstarved: n = 4; *thyA*$^-$ unstarved + H$_2$0$_2$: n = 5; *thyA*$^-$: n = 3; *thyA*$^-$+ H$_2$0$_2$: n = 3.

*Flow cytometry for the gentamicin assay for WT cells stained with Peroxy Orange*: After the pregrowth and washes as outlined above, *thyA*$^+$ cells were stained with Peroxy Orange dye as outlined above. Treated cells were also supplemented with 1μg/mL gentamicin for the lower dose ([Fig 4K]) or 4μg/mL gentamicin for the higher dose ([S14 Fig]). At 1.5h treatment, ROS accumulation was visualized using flow cytometry. The no drug cells were prepared, stained and visualized exactly as the drug-treated cells except for the addition of gentamicin. The number of independent experiments for visualization at 1.5h: *thyA*$^+$: n = 5; *thyA*$^+$+ gent(low): n = 3; t*hyA*$^+$+ gent(high): n = 2.

*FACS-based assessment of the relationship between cytoplasmic acidification and cell survival*: *thyA*$^-$ cells (in the MG1655 background) were grown overnight in high thymidine rich defined media and diluted 1 to 400 into high thymidine rich defined media for two hours of pregrowth in a 37°C shaking incubator. After two hours of pregrowth, cells were pelleted, washed in PBS, and the 2mL high thymidine rich defined media was replaced with 2mL zero thymidine rich defined media. Strains were stained with either BCECF or Phrodo green (the two pH sensitive dyes) in the last thirty minutes of the 1.5h thymidine starvation—exactly as was done for all the previous flow cytometry data (see above). After 1.5H of starvation at 37 degrees, an aliquot of 300uL was placed into a Bio-Rad S3e Cell Sorter. The first gate was defined to have a narrow FSC/SSC range that excluded 9/10th of the population so that the differences in cell volume across the cells sorted were negligible. From there, 1000 cells from

the top ~10% and 1000 cells from the bottom ~10% of the fluorescence distribution were plated on high thymidine LB agar plates and incubated at 37°C overnight to test for changes in viability on either end of the pH spectrum. 1000 cells were also sorted and plated for the top and bottom ~10% of the FITC fluorescence for a nodye control. We fitted the resulting count data with a Bayesian negative binomial regression with coefficients for the dye condition (including no-dye samples), interaction of dye with fluorescence level (high vs. low), and the date of the experiment. Models were fitted using brms [102] using default priors. The numbers of independent experimental replicates for sorting and survival assessment at 1.5h were: *thyA*⁻ stained with the pH sensitive dyes: BCECF: n = 5; Phrodo green: n = 7; *thyA*⁻ not stained and sorted based on the pH sensitive channel (FITC): n = 4.

*Controls to show that the pH dyes do not affect survival*: thyA⁻ cells were prepared exactly as above for the sorting experiment except that instead of sorting the cells at 1.5h, cells were plated on high thymidine plates and assessed for survival the following day. The survival of cells stained for these pH dyes is shown along with the survival of thyA- cells not stained at the same time point. The number of independent experiments for survival assessment at 1.5h: *thyA*⁻ stained with the pH sensitive dyes: BCECF: n = 5; Phrodo green: n = 4; *thyA*⁻ not stained: n = 12.

## Calculating MICs for gentamicin

The MIC for *thyA*⁻ (parental) and *ΔackA* were calculated using the Liofilchem MIC Test Strip for Gentamicin (cat. Number 22-777-677; 0.016–256 µg/mL) on LB plates supplemented with thymidine.

## Supporting information

**S1 Fig. Survival profiling uncovers known and novel contributors to TLD.** (A) Survival profiling time points, with and without outgrowths (see [Methods]). At 0h starvation and 3h starvation, an aliquot of the transposon library was placed in high thymidine for 2–3 doublings in order to amplify signal from living cells over residual signal from dead cells. An aliquot was also taken at the start of the selection, before the 0h outgrowth. (B) The distribution of survival scores and generation of candidate lists. 493 genes were above a threshold $\log_2$ fold change magnitude (shown by red dashed line) and had a *q*-value<0.05. Significance was calculated on the ratio of FPM at 3h vs. 0h using the rate ratio test and was corrected using the Benjamini & Yekutieli method. Any genes on this list that had an opposite sign from the survival score for the 3h vs. "Start" were discarded, yielding a total of 387 candidate genes. (C) is an inset for the area pictured in the rectangle in panel (B). (D) Survival scores (and significance) for genes previously known to have effects on TLD survival when knocked out. (E; G-H) Transposon insertions visualized using the Integrated Genome Browser. The frequency of each insertion site is shown in read counts per million. (E) Frequency of transposon insertions along the length of two previously known TLD contributors at 0h and after 3h thymidine starvation. A deletion in *uvrD* is known to sensitize cells to TLD and a deletion in *recO* is known to alleviate TLD. (F) Candidates were validated by transferring Keio collection knockout alleles into MG1655 *thyA*⁻, and assessing their survival at 3h of thymidine starvation. All death assays, unless otherwise stated, were performed at 37°C. Relative survival was measured for at least three independent experiments, with error bars representing standard error of the mean. *p*-values for all death assays were calculated using a Welch t-test. * P<0.05, ** P<0.01, *** P < 0.001, **** P<0.0001. (G) Enhancements of transposon insertions within the *cydAB* operon. (H) Enhancements of transposon insertions within the *speAB* operon. (TIFF)

**S2 Fig. The correlation of growth rate in high thymidine rich defined media vs. relative survival at 3h in zero thymidine media.** A. All strains. B. All strains minus the *ackA* strains. C. *ackA* strains only. See Methods under Growth Rate and Death Assays.
(TIFF)

**S3 Fig. Transcriptional responses of TLD-sensitive and TLD-resistant strains.** (A) Experimental setup of RNA sequencing of parental and evolved strains in the MDS42 background. All steps were performed at 30°C (the temperature at which laboratory evolution was conducted). Pregrowth was for 2h, and RNA was collected both before pelleting (1) and 30 minutes after pelleting and transfer into thymidine-free media (2). RNA was also collected 30 minutes after pelleting and placement back in high thymidine as a control (3). (B-C) $\log_2$ expression of genes in two comparisons showing significant differentially expressed genes in red. The 17 and 25 significant differentially expressed genes can be found in S4D and S4C Fig, respectively. (B) Parental starved vs. unstarved. (C) Evolved starved vs. unstarved.
(TIFF)

**S4 Fig. Gene-level analysis of transcriptional responses.** (A-D) Significant differentially expressed genes in various comparisons. See S8 Table for the *q*-values. The LFCs of RPKMs are visualized here. (A) Significant differentially expressed genes in the evolved vs. the parental in the unstarved condition. (B) Significant differentially expressed genes in the evolved vs. parental 30 minutes into thymidine starvation. (C) Significant differentially expressed genes in the evolved starved vs. evolved unstarved. (D) Significant differentially expressed genes in the parental starved vs. parental unstarved. As in the main text, purple represents genes involved in putrescine / glutamate / arginine metabolism or their associated amino acid decarboxylation acid resistance systems; orange represents genes involved in ROS; red represents genes involved in acetate dissimilation; and blue represents genes involved in DNA replication/ repair.
(TIFF)

**S5 Fig. Enrichment in genes with concordant responses vs. non-concordant responses between the survival profiling and transcriptome profiling experiments.** Genes that decrease survival when disrupted (with survival scores below -0.5) and are upregulated in the resistant strain (with $\log_2$ fold change TPM over 0.5) were enriched in all four RNA seq expression comparisons: (A) shows survival scores with the expression of resistant (evolved) starved vs sensitive (parental) starved; (B) shows survival scores with the expression of resistant (evolved) vs sensitive (parental) unstarved; (C) shows survival scores with the expression of resistant (evolved) starved vs resistant (evolved) unstarved; and (D) shows survival scores with the expression of sensitive (parental) starved vs sensitive (parental) unstarved.
(EPS)

**S6 Fig. Genes in DNA replication/repair, Electron transport chain and/or ROS accumulation, and pH homeostasis pathways with concordant genetic and transcriptional effects across experimental approaches.** (A-B) The left-hand column shows survival scores from the survival profiling experiment and the right-hand column shows LFC of mRNA expression (TPM) from the transcriptome profiling experiment. (A) Survival scores with the LFC of RNA expression in evolved starved vs. evolved unstarved. The remaining genes showing concordant effects can be found in S10 Table. (B) Survival scores with the LFC of RNA expression in evolved unstarved vs. parental unstarved. The remaining genes showing concordant effects can be found in S11 Table. Colors are used to represent recurring pathways as described

in S6 Fig. Pink represents genes involved in proton translocation (or sequestration) systems or in the acid stress response.
(TIFF)

**S7 Fig. Kill curves for the most sensitive strains before, at, and after the chosen flow cytometry time points.** The killing process of the *thyA*⁻ starting strain in the MG1655 background is shown in black and in the MDS42 background is shown in red for the first 5h of thymidine starvation as well as at 24h thymidine starvation. The 1.5h and 3h time points were chosen for flow cytometry experiments.
(EPS)

**S8 Fig. Parental strains in MG1655 and MDS42 backgrounds show evidence of ROS accumulation and acidification at 3h thymidine starvation while the evolved TLD-resistant strains do not.** (A-C) Adjusted fluorescence measured using flow cytometry. All strains are in thymidine-free media. (A) Adjusted fluorescence measured at 3h for strains dyed for hydrogen peroxide using Peroxy Orange. (B) Adjusted fluorescence measured at 3h for strains dyed for pHi using PHrodo Green. (C) Adjusted fluorescence measured at 3h for strains dyed for pHi using BCECF-AM. See Fig 3 caption for definitions of plotted intervals and significance tests.
(TIFF)

**S9 Fig. *thyA*⁻ strain shows evidence of ROS accumulation at 1.5h only when exogenous $H_2O_2$ is added.** Adjusted fluorescence measured using flow cytometry of *thyA*⁻ cells stained with Peroxy Orange with and without a 30-minute incubation with 1mM $H_2O_2$. The two plots on the left show adjusted fluorescence at 1.5h in high thymidine media and the two plots on the right show adjusted fluorescence at 1.5h thymidine starvation. See Fig 3 caption for definitions of plotted intervals and significance tests.
(TIFF)

**S10 Fig. Fluorescence activated cell sorting shows that cytoplasmic acidification is associated with lower survival at early timepoints during TLD.** (A) *thyA*⁻ cells starved of thymidine for 1.5h and stained with one of two pH sensitive dyes for the last half an hour of starvation were sorted into the top and bottom 10% of the fluorescence distribution and then plated onto thymidine-supplemented plates to assess for changes in survival. Shown are the fitted ratios of survival rates from the lower acidification sorted population versus the rates for survivors from the high acidification sort. The nodye control is sorted and plated exactly like the test but without any staining in the last half hour of starvation. (B) *thyA*⁻ cells starved of thymidine for 1.5h and stained with one of two pH sensitive dyes for the last half an hour of starvation were plated onto thymidine supplemented plates to confirm that the dyes do not cause significant changes in survival.
(EPS)

**S11 Fig. The *ΔackAΔcydB* double mutant does not show increased survival.** Survival of the parental strain and knockouts in the MG1655 background at 3h thymidine starvation. Relative survival was measured for at least three independent experiments, with error bars representing standard error of the mean.
(TIFF)

**S12 Fig. Growth at 30 degrees has less effect alleviating TLD than the *ackA* and the *recO+ackA* mutants.** Survival of the parental strain (*thyA*⁻) and knockouts in the MG1655 background at 3h thymidine starvation at 37° vs 30°. Relative survival was measured for at least three independent experiments, with error bars representing standard error of the mean. T-test significance for *ΔackA* vs. *ΔackAΔrecO* = 8.24 x 10⁻⁰⁵ and for the *thyA*⁻ at 30° vs. *ΔackA* = 5.29 x 10⁻⁴.
(TIFF)

**S13 Fig. Viability 1.5h into drug treatment.** *thyA⁺* cells were treated with 1μg/mL gentamicin and plated for colony counting on LB/thymidine plates at 0h and again after 1.5h of shaking in defined media at 37°C.
(EPS)

**S14 Fig. At higher concentrations of gentamicin, both acidification and ROS accumulation are observed.** Plots of adjusted fluorescence of wild type (*thyA⁺*) cells measured by flow cytometry at 2h with and without 1μg/mL or 4μg/mL gentamicin using pHrodo Green for pHi (A) or Peroxy Orange for ROS (B). See Fig 3 caption for definitions of plotted intervals and significance tests.
(TIFF)

**S1 Table. Survival scores and significance for survival profiling of transposon library generated in MG1655 thyA⁻ genetic background.**
(XLSX)

**S2 Table. 212 candidate genes in which insertions exacerbate killing.**
(XLSX)

**S3 Table. 175 candidate genes in which insertions alleviate killing.**
(XLSX)

**S4 Table. List of genes previously known to have effects on TLD and the survival scores from our Survival Profiling screen.**
(PDF)

**S5 Table. List of 52 candidates with positive survival scores to validate.**
(XLSX)

**S6 Table. Mutations in the TLD-resistant evolved isolate in the MG1655 background.**
(XLSX)

**S7 Table. TPMs of RNA expression of parental and evolved strains in the MDS42 genetic background.**
(XLS)

**S8 Table. RPKM, TPMs and *q*-values of significant differentially expressed genes.**
(XLSX)

**S9 Table. Candidate genes that show concordant effects in genetic and transcriptional responses in evolved starved vs. parental starved.**
(XLSX)

**S10 Table. Candidate genes that show concordant effects in genetic and transcriptional responses in evolved starved vs. evolved unstarved.**
(XLSX)

**S11 Table. Candidate genes that show concordant effects in genetic and transcriptional responses in evolved unstarved vs. parental unstarved.**
(XLSX)

## Acknowledgments

We wish to thank past and present members of the Tavazoie laboratory for critical discussion and comments on the manuscript. We are also grateful to Siu-Hong Ho and Wei Wang at the

Columbia University Herbert Irving Comprehensive Cancer Center Flow Cytometry Core for help using the Fortessa instrument.

## Author contributions

**Conceptualization:** Alexandra Ketcham, Saeed Tavazoie.

**Data curation:** Alexandra Ketcham.

**Formal analysis:** Alexandra Ketcham, Lydia Freddolino, Saeed Tavazoie.

**Funding acquisition:** Saeed Tavazoie.

**Investigation:** Alexandra Ketcham, Lydia Freddolino.

**Methodology:** Alexandra Ketcham, Lydia Freddolino.

**Project administration:** Saeed Tavazoie.

**Resources:** Saeed Tavazoie.

**Supervision:** Lydia Freddolino, Saeed Tavazoie.

**Validation:** Alexandra Ketcham, Lydia Freddolino, Saeed Tavazoie.

**Visualization:** Alexandra Ketcham, Lydia Freddolino.

**Writing – original draft:** Alexandra Ketcham, Lydia Freddolino, Saeed Tavazoie.

**Writing – review & editing:** Alexandra Ketcham, Lydia Freddolino, Saeed Tavazoie.

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
