## [Decision Letter · Decision Letter 0]

20 May 2022

Dear Dr Tavazoie,

Thank you very much for submitting your Research Article entitled 'Intracellular acidification is a hallmark of thymineless death in E. coli' to PLOS Genetics.

The manuscript was fully evaluated at the editorial level and by independent peer reviewers. The reviewers appreciated the attention to an important problem, but raised some substantial concerns about the current manuscript. You will notice that the concerns of the reviewers overlapped (use of clear language, clarity of figures, interpretation of old data on TLD including related to ROS, etc), and that reviewer #2 has raised serious concerns about the Tn-seq library and procedure used in the study as well as the paucity of statistical support for inferences drawn. Based on the reviews, we will not be able to accept this version of the manuscript, but we would be willing to review a much-revised version. We cannot, of course, promise publication at that time.

If you decide to revise the manuscript for further consideration at PLOS Genetics, please aim to resubmit within the next 60 days, unless it will take extra time to address the concerns of the reviewers, in which case we would appreciate an expected resubmission date by email to plosgenetics@plos.org .

If present, accompanying reviewer attachments are included with this email; please notify the journal office if any appear to be missing. They will also be available for download from the link below. You can use this link to log into the system when you are ready to submit a revised version, having first consulted our Submission Checklist .

While revising your submission, please upload your figure files to the Preflight Analysis and Conversion Engine  (PACE) digital diagnostic tool.  PACE helps ensure that figures meet PLOS requirements. To use PACE, you must first register as a user. Then, login and navigate to the UPLOAD tab, where you will find detailed instructions on how to use the tool. If you encounter any issues or have any questions when using PACE, please email us at figures@plos.org .

PLOS has incorporated Similarity Check , powered by iThenticate, into its journal-wide submission system in order to screen submitted content for originality before publication. Each PLOS journal undertakes screening on a proportion of submitted articles. You will be contacted if needed following the screening process.

[LINK]

We are sorry that we cannot be more positive about your manuscript at this stage. Please do not hesitate to contact us if you have any concerns or questions.

Yours sincerely,

Diarmaid Hughes

Associate Editor

PLOS Genetics

Lotte Søgaard-Andersen

Section Editor: Prokaryotic Genetics

PLOS Genetics

Reviewer's Responses to Questions

**Comments to the Authors:**

Reviewer #1: The study uses three complementary approaches to examine the TLD phenomenon in E. coli:

1. Tn-Seq profiling to find genes that enhance survival during thymidine starvation

2. Laboratory evolution to find mutants with TLD resistance

3. Transcriptomics from thymidine-starved and non-starved ancestral and evolved strains to identify genes that are differently affected by thymidine starvation in the evolved vs. the ancestral strains

Aside from identifying genes known from previous studies to be involved in TLD, the study identifies genes directly or indirectly involved in pH homeostasis as part of TLD. The data from the Tn-Seq, evolution experiments, and transcriptomics are solid and complements each other very nicely.

I have no major concerns, and only two comments that I think could improve the manuscript:

Comment #1: The use of flow cytometry to measure fluorescence of bacteria has been shown to be prone to artefacts due to changes in cells size caused by e.g. treatment with various antibiotics. This has been addressed and is described in the Methods section, but I think it could also be mentioned in appropriate places in the results section (and perhaps discussed as part of addressing Comment #2).

Comment #2: The involvement of ROS in cell death during TLD is controversial (and connected to the even more controversial hypothesis that ROS is involved in a unifying killing mechanism shared by several different classes of bacteriocidal antibiotics). The ROS damage hypothesis in TLD was originally supported by experiments where the cells were treated with the Fe(II) chelator bipyridyl to block oxidative damage caused by Fenton’s reaction. Bipiridyl prevented TLD and prevented the detection of ROS (as detected by flow cytometry in dye-labeled cells without correction for changes in cell size). However, more careful experiments indicated that the concentration of bipyridyl that was used actually inhibited growth, protecting against TLD by preventing DNA replication, and preventing the increase in fluorescence intensity of the dye-labeled cells by preventing cell growth. Use of another Fe(II) chelator that did not inhibit growth did not prevent TLD or increase ROS detection. An even more compelling blow against the ROS hypothesis in TLD was that adding exogeneous peroxide instead of aggravating TLD actually protected cells from TLD (for reference: https://journals.asm.org/doi/10.1128/JB.00370-21 ). I suggest the authors add a paragraph about this controversy to the discussion, with their arguments for why the ROS detection and pHi measurements reflect real changes in cell physiology that are part of the TLD phenomenon (perhaps including something about the risk of artefacts due to changes in cell size and shape and how to avoid it).

Reviewer #2: Thymineless death is an important phenomenon, resulting in rapid cell death in E. coli and other organisms. It has a long literature but lacks a complete understanding of its mechanistic underpinnings. In their manuscript “Intracellular acidification is a hallmark of thymineless death in E. coli”, Ketcham, Freddolino, and Tavazoie use three genome scale technologies to try to dissect key additional features of the response, suggesting that intracellular acidification is a causal factor in thymineless death. Unfortunately, there are issues with their experiments, and additional experimentation and analyses are required to solidify their claims, as detailed below

1. The authors perform a Tn-seq screen to determine the effect of various E. coli knockouts on survival of a delta-thyA strain during thymidine starvation. I have concerns with both how the experiment was performed and with the data analysis.

a. The authors use a non-standard protocol for their Tn-seq experiment. Rather than collecting a sample at the onset of thymidine starvation and after 3 hrs of starvation, the authors collect these samples only after an additional 3 hrs of growth in high thymidine. Therefore, “hits” are those cells that resist TLD, and those cells that recover more rapidly. These may be different gene classes, perhaps explaining the plethora of hits.

The authors justify this non-standard protocol by saying that this will “amplify signal from live cells relative to any residual DNA from dead cells”. If necessary, the authors could take several timepoints after initiation of thymineless death to determine which mutants are resistant, and then take timepoints after re-addition of thymidine to make a further determination of which mutants show more rapid recovery.

b. Additionally, taking multiple timepoints of the Tn-seq library under thymidine starvation without outgrowth (e.g. 1hr, 2hr, etc) would help the authors make many of their claims much more cleanly. For example, they could look at the length of time before the onset of TLD in addition to the rate of TLD in different classes of genes.

Performing the Tn-seq screen in multiple ways (i.e. marginal thymidine concentration, drugs, etc.) would also provide additional insight and decrease the number of false positives they find.

c. Was there a single replicate of the screen performed? The authors should at least do technical replicates of the screen to quantify the variability in their experimental protocol.

d. Were transposon insertions at the ends of genes considered in the analysis? It is customary to disregard the first and last 5% or so of a gene, because transposon insertions in these locations often don’t inactivate the gene.

e. Why are ileY and trpL the strongest significant positive and negative significant hits? If these small genes are spurious hits (and it certainly seems that way), the authors should re-evaluate how they do their Tn-seq analysis so that these don’t get marked as significant.

2. Given the very large number of genes (~400) that the authors identified as having “significant” phenotypes in their Tn-seq screen, a rigorous statistical treatment is needed to support the inferences drawn by the authors. This is a major issue with the manuscript as written! A few examples below:

a. “As expected, disruptions in genes whose inactivation have previously been shown to sensitize cells to TLD, such as uvrD [21-23], have significant negative survival scores” Do genes that have been previously implicated as important for surviving TLD have overall negative scores?

b. “…disruptions in genes whose inactivation have previously been shown to enhance survival, such as recO [22, 24, 25], have significant positive survival scores (Figs 1B-C, S1D Fig).” Again, is this generally true?

c. “Twenty additional genes with significant fitness effects fall in the previously known pathways of DNA replication and repair, and respiration (Figs 1B-C; Table 1).” Are genes involved in respiration and/or DNA replication and repair significantly enriched in these sets? If one were to pick 400 genes at random some would undoubtably be involved in respiration and DNA processes.

d. This issue is most pronounced when the authors discuss their most important finding – that genes involved in “the newly identified pathway of pH homeostasis” show TLD phenotypes. There is absolutely no statistical support for this in the manuscript. Where is the data that shows that “genes that produce or import H+ into the cytoplasm, or that lower levels of substrates needed for deacidification systems, enhance survival during thymidine starvation”? Or that “disruptions in genes that consume protons, or produce substrates needed for deacidification systems, exacerbate killing”? Table 2 lists some genes that fall into these categories. But the authors have ~400 genes. There are an unlimited number or “stories” that one could tell by cherry-picking data like this.

3. The authors are clearly aware of the literature surrounding TLD in E. coli, and they do an excellent job of summarizing it in the introduction. However, in their analysis they fail to consider previous explanations for their observations. For example, since ROS response and respiration have been shown to play a role in TLD resistance or susceptibility, a novel gene should first be demonstrated not to function by affecting these pathways before a new category of function is proposed. For example, is it possible that ackA plays a role in respiration (or affects its regulation), rather than pH homeostasis?

I suggest that the authors revisit and integrate the large body of previous E. coli screens to help contextualize their data. For example, Nichols et al., 2011 (10.1016/j.cell.2010.11.052) performed screens of the E. coli Keio collection under ~300 conditions, including azidothymidine, pH4, and trimethoprim. Interestingly, ackA was the top surviving strain under high azidothymidine.

4. The authors perform an evolution experiment on two E. coli strains to bolster their findings. I have a two major concerns about this experiment and its interpretation.

First, why were only 2 strains sequenced/discussed in the main text? The authors collected 6 isolated from each of 4 plates in 2 backgrounds. Why not at least sequence one strain from each of the 4 replicates in their experiment? This would give the authors a way to filter random mutations from significant ones and would bolster the credibility of their results.

Second, the authors should be much more clear about what MSD42 is: a MG1655 strain with 699 genes deleted. This hardly justifies the authors should walk back their verbiage on “diverse strains” in the abstract and elsewhere. Out of curiosity, why was this strain chosen instead of something less engineered and more diverse? I personally worry about results from this strain, as many “minimal genome” strains have regulatory loops engineered out of them. Therefore, any results in this strain must be shown to be significant in the wild-type strain as well.

5. The authors perform RNA-seq on the evolved strains under thymine starvation and then focus on genes with consistent responses – those that are upregulated and the KO is harmful or vice-versa. Is there an enrichment in genes with “consistent” responses vs non-consistent responses? If so, this would support the logic of the authors’ strategy. If not, the analysis does not make sense. Also, many previous studies have found little relationship between expression and fitness.

6. The authors’ analysis of pHi is intriguing, but I feel like it does not answer the question of whether pHi is causal or not.

a. Do all (or a majority) of genes that confer acid resistance also confer resistance to TLD?

b. Fig 3 and Fig 4 are extremely difficult to read. Can the authors convert the fluorescence values to pH values? This would both make the figures easy to interpret and would give biologists a sense of how large the reported changes are.

c. Since the authors use a fluorescent probe of cytoplasmic pH why is the temporal resolution so bad? The authors should show cytoplasmic pH on a scale of minutes as a function of thymidine starvation. This can be done either using live cell microscopy or flow cytometry. This would relieve a major concern, namely, pH changes occur after >90% killing, as judged by the literature. If the authors kill curves mimic literature findings, pH change cannot be causal

d. An additional strategy that the authors could use to show the importance of acid stress response is to use FACS to sort an isogenic population of thymine starved cells into those with high and low cytoplasmic pH and demonstrate that survival differs between those populations.

Minor points:

1. L43- “mapped and sequenced at different time points” —as far as I can tell the authors only took one timepoint (3hrs).

2. Fig 1G: L81—were these clean deletions or antibiotic replacements which have an outwardly facing promoter to express downstream gene?

3. When discussing the evolution experiments, I think it is important to mention that for the majority of the transfers (43/50) only 2-fold dilutions were performed.

4. In Fig 3, the “+” on thyA+ is extremely small and difficult to read. Perhaps you could replace thyA+ with WT?

5. Fig 3 Why are the values for the same strains/conditions not consistent across panels? For example, thyA+ is ~4 in panel E and ~0 in panels F, L, and M, despite having the same Y-axis label.

Reviewer #3: “Intracellular acidification is a hallmark of thymineless death in E. coli” by A. Ketcham et al. (PLOS Genetics).

In this work the authors investigate the phenomenon of thymineless death (TLD), in which E. coli cells that cannot synthesize the essential compound thymine (thyA cells) are rapidly inactivated (i.e., they die) when asked to grow in a medium lacking either thymine or thymidine. This is in contrast to starvation for most other essential growth components, where the starved cells simply stop growing and enter stasis rather than suddenly dying. This TLD phenomenon has been studied for many decades, and many theories have been proposed to account for the precise mechanism underlying this type of cell death. However, none has appeared to fully account for the TLD. Here, the authors find evidence for yet another mechanistic aspect associated with TLD, namely intracellular acidification, which appears rather early in the death process and may be a major contributing cause to the cell death. The discovery is based on several experimental findings: 1. A saturating search for gene knockouts that either improve or exacerbate TLD reveals several hits of genes controlling cell acidification. 2. The experimental development of TLD-resistant thyA strains shows that these adapted cells have many mutations among which intracellular pH genes are prominent. 3. mRNA expression studies revealing that acidification control genes are specifically affected during TLD. 4. Experimental manipulation of the growth medium that is expected to prevent intracellular acidification (addition of the amino acid arginine) does indeed prevent or diminish TLD. 5. Fluorescent dye indicators are use to directly demonstrate the experimental association of TLD with acidification.

Overall, this is very interesting work. While intracellular acidification is unlikely to be final chapter in understanding TLD, it seems to me that this is an important addition to our current knowledge on the subject. The work seems well performed and interpreted. Publication of this work is recommended.

Here are some issues that the authors need to address:

1. While the Methods section and Figure Legends are chock-full of experimental details (good), the main text suffers as it is in places too succinct and lacks sufficient information for the average reader who may not be fully conversant with all the experimental approaches and lab jargon. The main text should tell a story that can be easily followed by the interested reader, without having to frantically go back and forth between text, Figure Legends, Methods, supplementary Figs and Tables and their associated Legends (as happened to this reviewer, and this is painful). Please improve and clarify your main narrative.

As one example, let’s look at the beginning of the Results section. Here it reads: “A saturated Tn5 transposon insertion library generated in the thyA strain was selected in thymidine-free media and insertions were mapped and sequenced at different time points to look for changes in survival caused by the gene disruptions (Fig. 1A)”. This is a solid experimental approach, but the description is not clear. First of all, the authors did not select an insertion library…. The proper description is that they first created an insertion library (see Methods) and subsequently used that library to select mutants affected in TLD. This is a point of confusion, and a clearer description could have saved this reviewer a lot of time. Secondly, when you talk about changes in survival, you need to state (briefly) how you assess survival. Again, unfortunately I had to spend unnecessary time perusing all the details in Methods and Figure Legends to see how you did it. One or two simple sentences in the main text would help the flow. Note that there is no survival curve in the referred to Fig. 1A. I know from previous workers in this field that the survival curves in TLD are somewhat unusual and may contain several informative phases, and the interested reader would like to see what you are talking about.

2. Then the text continues: 'A survival score for each gene was defined as the log2 fold change (LFC) of fragments per million (FPM) at 3 hours (h) versus FPM at 0h". You need to describe this much better. Maybe if you showed some survival curves or reproduced some survival fractions you could guide the reader to how you get to your "survival score". You must make sure that the reader understands this before going on with the rest of the paper. Also, I have no idea what you mean by "fragments per million". What are these fragments? I am sure you are not studying DNA fragmentation, as some have done? The reader cannot go on without being sure what you are talking about. Also, describe what you mean by the positive and negative values for survival scores in the Tables 1 and 2. One could guess, but the authors should be more helpful at this point.

3. With regard to the data in Table 3 describing the listing the altered genes in the evolved strains that appear adapted to TLD, there is a discrepancy with the surrounding text. The text describes mutations in oriC (line 100), but I do not see this in the Table. Correct or explain.

4. The description of the genesis of the evolved strains could also use clarification. From the Methods I learn that for each strain (MG1655 and MDS42) four independent lines were started and processed for 50 daily transfer cycles. After 50 transfers the eight cultures were tested for survival, and the best surviving were spread on thymidine-containing medium (line 148). What is meant here? Did you pick the best survivor from MG1655 and the best survivor from MDS42 (ie, one for each strain)? Please be more clear. Then, you picked six colonies from each best survivor and retested them individually (line 154). From these six, the two top candidates were discussed in the main text (lines 153-155). So, to be clear, the two top candidates discussed for each strain were true siblings, ie, they came from exactly the same evolved culture. This was not clear from the first reading and caused unnecessary confusion. It is important to state this fact separately because it appears that the two top candidates of strain MDS42 shared exactly the same mutations (line 93) except for the atpF gene (line 103). Of course if the two sequenced candidates had been selected from two independent evolved lines this would have been highly unusual.

5. The first listed mutation in Table 3 (atpF gene) is listed as Q85*. Does this * refer to some footnote that I did not see? Is this the one with the stop codon? If so, what is the other one? (As an aside, it is interesting that this evolved culture contained two different atpF mutants, ie, the culture was heterogeneous, despite the fact that one survives significantly better than the other, see Fig. 2D).

6. I would like to see some more description and explanation of the interesting Figs. 2F and 2G. Can I assume that the 45-degree line represents all the transcripts that do not change in the given comparison? Does that also mean that the colored genes represented outside the 45° lines are the only ones that fall outside of this line, or do you list only those genes that you are interested in? (I hope not). But provide a better narrative for this.

7. Likewise, Figs. 2H and 2I are interesting but could also use some better explanation. See point 1 above regarding the survival score and FPM, but we also need a better description of the RNA part: LFC of RNA expression (TPM) (transcripts per million?).. A lot of jargon, please explain.

8. Starting on line 173, the authors start using the word "consistent". This is somewhat confusing. I most people's mind this word would mean "reproducible". However, I now realize that the authors mean to say that the effects of genetic modulation and transcriptional responses move in the same expected direction, i.e., consistent with expectation. I suggest other wording, like "corresponding" or perhaps define 'consistent' as 'consistent with expectation' the first few times it is used.

**Have all data underlying the figures and results presented in the manuscript been provided?**

Reviewer #1: Yes

Reviewer #2: Yes

Reviewer #3: Yes

PLOS authors have the option to publish the peer review history of their article (what does this mean? ). If published, this will include your full peer review and any attached files.

**Do you want your identity to be public for this peer review?** For information about this choice, including consent withdrawal, please see our Privacy Policy .

Reviewer #1: No

Reviewer #2: No

Reviewer #3: No

---

## [Decision Letter · Decision Letter 1]

1 Oct 2022

Dear Dr Tavazoie,

We are pleased to inform you that your manuscript entitled "Intracellular acidification is a hallmark of thymineless death in E. coli" has been editorially accepted for publication in PLOS Genetics. Congratulations!

Once your paper is formally accepted, an uncorrected proof of your manuscript will be published online ahead of the final version, unless you’ve already opted out via the online submission form. If, for any reason, you do not want an earlier version of your manuscript published online or are unsure if you have already indicated as such, please let the journal staff know immediately at plosgenetics@plos.org .

In the meantime, please log into Editorial Manager at https://www.editorialmanager.com/pgenetics/ , click the "Update My Information" link at the top of the page, and update your user information to ensure an efficient production and billing process. Note that PLOS requires an ORCID iD for all corresponding authors. Therefore, please ensure that you have an ORCID iD and that it is validated in Editorial Manager. To do this, go to ‘Update my Information’ (in the upper left-hand corner of the main menu), and click on the Fetch/Validate link next to the ORCID field.  This will take you to the ORCID site and allow you to create a new iD or authenticate a pre-existing iD in Editorial Manager.

Yours sincerely,

Diarmaid Hughes

Academic Editor

PLOS Genetics

Lotte Søgaard-Andersen

Section Editor

PLOS Genetics

Comments from the reviewers (if applicable):

Reviewer's Responses to Questions

**Comments to the Authors:**

Reviewer #1: Dear authors,

I am happy with your response and the changes you made based on my comments, and I also think the changes you made due to the concerns and comments from the other two referees have improved the manuscript significantly.

Reviewer #3: Sufficient modifications have been introduced to make the work more accessible and convincing.

**Have all data underlying the figures and results presented in the manuscript been provided?**

Reviewer #1: Yes

Reviewer #3: Yes

PLOS authors have the option to publish the peer review history of their article (what does this mean? ). If published, this will include your full peer review and any attached files.

**Do you want your identity to be public for this peer review?** For information about this choice, including consent withdrawal, please see our Privacy Policy .

Reviewer #1: No

Reviewer #3: No

**Data Deposition**

http://datadryad.org/submit?journalID=pgenetics&manu=PGENETICS-D-22-00458R1

More information about depositing data in Dryad is available at http://www.datadryad.org/depositing . If you experience any difficulties in submitting your data, please contact help@datadryad.org for support.

**Press Queries**

---

## [Editor Report · Acceptance letter]

17 Oct 2022

PGENETICS-D-22-00458R1

Intracellular acidification is a hallmark of thymineless death in E. coli

Dear Dr Tavazoie,

We are pleased to inform you that your manuscript entitled "Intracellular acidification is a hallmark of thymineless death in E. coli" has been formally accepted for publication in PLOS Genetics! Your manuscript is now with our production department and you will be notified of the publication date in due course.

With kind regards,

Zsofia Freund

PLOS Genetics

On behalf of:
